



**Assessment of Arctic and Antarctic Sea Ice Predictability in CMIP5 Decadal Hindcasts**
Chao-Yuan Yang[1], Jiping Liu[1], Yongyun Hu[2], Radley M. Horton[3], Liqi Chen[4], Xiao Cheng[5]
[1]Department of Atmospheric and Environmental Sciences, University at Albany, State University
of New York, Albany, NY, USA
[2]Department of Atmospheric and Oceanic Sciences, School of Physics, Peking University,
Beijing, China
[3]Columbia University Center for Climate Systems Research and NASA Goddard Institute for
Space Studies, New York, NY, USA
[4]Key Laboratory of Global Change and Marine-Atmospheric Chemistry, Third Institute of
Oceanography, SOA, Xiamen, China
[5]College of Global Change and Earth System Science, Beijing Normal University, Beijing, China
Corresponding author:
Chao-Yuan Yang
Department of Atmospheric and Environmental Sciences
University at Albany, State University of New York
Albany, NY, 12222 USA
Email: cyang4@albany.edu



**Abstract**
This paper examines the ability of coupled global climate models to predict decadal
variability of Arctic and Antarctic sea ice. We analyze decadal hindcasts/predictions of 11
CMIP5 models. Decadal hindcasts exhibit a large multi-model spread in the simulated sea ice
extent, with some models deviating significantly from the observations. For the models having
large biases and using full-field initialization, the predicted sea ice extent quickly drifts away
from the initial constraint, deteriorating the decadal predictive skill. The anomaly correlation
analysis between the decadal hindcast and observed sea ice suggests that in the Arctic, for most
models, the areas showing significant predictive skill become broader associated with increasing
lead times. This area expansion is largely because nearly all the models are capable of predicting
the observed decreasing Arctic sea ice cover. Sea ice extent in the north Pacific has better
predictive skill than that in the north Atlantic (particularly at a lead-time of 3-7 years), but there
is a re-emerging predictive skill in the north Atlantic at a lead-time of 6-8 years. In contrast to
the Arctic, Antarctic sea ice decadal hindcasts do not show broad predictive skill at any time
scales, and there is no obvious improvement linking the areal extent of significant predictive skill
to lead-time increase. This might be because nearly all the models predict a retreating Antarctic
sea ice cover, opposite to the observations. For the Arctic, the predictive skill of the MMEE
outperforms most models and the persistence prediction at longer time scales, which is not the
case for the Antarctic.



## 1. Introduction

Decadal climate prediction is a new and rapidly evolving research area driven by societal demand for climate information to inform climate adaptation strategies (e.g., Meehl et al., 2009, 2013; Vera et al., 2010). As a boundary between the ocean and atmosphere, sea ice plays an important role in the climate system and acts as an important indicator of climate change through dynamic and thermodynamic processes and various feedbacks (i.e., albedo, insulation and buoyancy). Thus, sea ice simulation and prediction is one of the most challenging and important issues in decadal climate prediction, i.e., Meehl et al. (2009) emphasized the importance of sea ice treatment in climate models as large uncertainties remain for decadal climate prediction.

In the past few decades, Arctic sea ice has been declining (e.g., Serreze et al., 2007; Arctic Report Card, 2015). Trends in Arctic sea ice extent are negative for all months (e.g., Comiso, 2008, 2012; Cavalieri and Parkinson, 2012) largely due to thinning and loss of the perennial sea ice cover (Kwok et al., 2009), but are largest at the end of the summer melt season. September Arctic sea ice extent has declined by $0.87 \times 10^6$ km$^2$ for the period 1979-2014, with a pronounced decreasing trend of sea ice concentrations in the arc extending from the Beaufort Sea to the Barents Sea (> 95% significance, Fig. 1a). The possibility of an ice-free Arctic in the coming decades (Stroeve et al., 2007, 2012; Boé et al., 2009; Wang and Overland, 2009, 2012; Zhang, 2010; Massonnet et al., 2012; Liu et al., 2013) would have profound impacts on Arctic maritime activities (e.g., opening of shorter shipping routes) and ecosystems (e.g., changing solar radiation in the upper ocean and influencing primary productivity), and extreme weather and climate in mid- and high-latitudes (e.g., Liu et al., 2012; Francis and Vavrus, 2012; Smith and Stephenson, 2013; WWRP/PPP, 2013; Stroeve et al., 2014).



By contrast, Antarctic sea ice has been expanding (e.g., Liu et al., 2004; Turner et al., 2009;
Comiso et al., 2011; Parkinson and Cavalieri, 2012). Trends in Antarctic sea ice extent are
positive for all months. Unlike the almost uniform Arctic sea ice decreases, the trends in
Antarctic sea ice concentrations show strong regional variations, although the NASA's Ice,
Cloud, and land Elevation Satellite showed that Antarctic sea ice thickness has a small negative
trend during 2003-2008 (Kurtz and Markus, 2012). September Antarctic sea ice extent has
increased by $0.24 \times 10^6$ km$^2$ per decade during 1979-2014, with a pronounced positive trend of
sea ice concentrations in the Ross Sea partially offset by a negative trend in the Bellingshausen
and Weddell Seas (Fig. 1b). The limited understanding of some of the mechanisms responsible
for the observed decrease (increase) in Arctic (Antarctic) sea ice makes sea ice prediction
challenging (e.g., Kattsov et al., 2010; Richter-Menge et al., 2012; Bindoff et al., 2013; Goosse
et al., 2015).
Most sea ice predictability studies have focused on the Arctic and the
seasonal-to-interannual time scale. An outlook of September Arctic sea ice extent has been
solicited from research community since 2008. Stroeve et al. (2014) showed that the median July
(the same was true for June and August) prediction value for September sea ice cover was off by
a large margin in 2009, 2012 (record low), and 2013. Koenigk and Mikolajewicz (2009)
suggested sea ice cover has low predictability in the central Arctic but some predictability at sea
ice edge zones in the MPI ECHAM5-OM climate model. Holland et al. (2011) showed potential
predictability of sea ice cover with a few months lead-time in the NCAR Community Climate
System Model version 3 (CCSM3). They also suggested that the persistence of sea ice thickness
anomalies is much higher than that of sea ice extent anomalies, which might point to a pathway
towards greater predictability as models improve their simulation of sea ice thickness.



Predictability of sea ice cover with e-folding time scales of 2-5 months has been identified in
several climate models (Day et al., 2014a). A few modeling studies also showed continuous
predictability of sea ice cover for 1-2 years, and intermittent predictability for 2-4 years
(Blanchard-Wrigglesworth et al., 2011b; Day et al., 2015; Tietsche et al., 2013, 2014; Guemas et
al., 2014). In contrast to the Arctic, there are limited efforts on examining predictability of
Antarctic sea ice. Using the NCAR CCSM3 model, Holland et al. (2013) showed initial-value
predictability of sea ice for a few months in the edge around Antarctica.
To date, relatively little attention has been paid to assessing prediction skill of sea ice at
decadal timescales for the Arctic and Antarctic in current-day climate models. Decadal sea ice
prediction entails a combination of initial value and climate forcing issues. At decadal timescales,
internal climate variability affects sea ice (i.e., some aspects of climate internal variability may
be predictable, Collins and Allen, 2002; Smith et al., 2007; Keenlyside et al., 2008; Meehl et al.,
2009; Pohlmann et al., 2009; Mochizuki et al., 2012), as does prescribed external scenarios (e.g.,
greenhouse gases and other radiatively important agents). Blanchard-Wriggleworth et al. (2011b)
suggested that predictability of Arctic sea ice beyond 3 years is largely influenced by climate
forcing rather than initial values. The growing dominance of climate forcings is likely to
introduce some potential predictability since it accounts for increasingly large portions of sea ice
change from present conditions (e.g., National Research Council, 2012). Guemas et al. (2014)
also underlined that predicting future change of Arctic sea ice on decadal timescales is
challenging due to initialization problems (i.e., the initial shocks due to sparse observations,
limitations of reanalysis data, and ensemble generation methods).
The recent Coupled Model Intercomparison Project Phase 5 (CMIP5) has implemented an
experiment to simulate and predict decadal climate variability (Meehl et al., 2009; Taylor et al.,





2012) in support of the Intergovernmental Panel in Climate Change Fifth Assessment Report.
The validation of decadal hindcasts is an important step for improving decadal predictions, since
it can elucidate issues in initialization methods and model responses to natural variability and
climate forcings. In this study, we examine the capability of CMIP5 decadal hindcasts to
simulate the mean and decadal variability of Arctic and Antarctic sea ice extent.
**2. Models and data**
Eleven CMIP5 models are used to evaluate the decadal hindcast/prediction of sea ice in both
the Arctic and Antarctic. These eleven models provide a set of 10-year long hindcast simulations,
which was initialized every five-years from 1981 to 2006. The purpose of initialization is to start
coupled global climate models close to the most realistic possible sea ice state. In general, the
initialization for the CMIP5 decadal hindcast/prediction can be divided into two approaches, full
initialization and anomaly initialization. For the full initialization approach, the initial model
state is replaced by the best available estimate of the observed sea ice state (i.e., satellite
observation and ocean analysis). This efficiently reduces the initial error due to the systematic
bias in the presence of model deficiencies. However, as the model is integrated for the decadal
hindcast/prediction, the simulation tends to drift away from the best-estimated sea ice state no
matter how small the initial error is. The anomaly initialization approach partly addresses this
problem by assimilating observed sea ice anomalies on the modeled sea ice state with focus on
predicting future sea ice anomalies.
Table 1 provides a summary of the initialization approaches and data source of the initial sea
ice state for each individual model. More detailed information about the set-up of the decadal
experiment can be found in Meehl et al. (2009) and Taylor et al. (2012). For each individual
model, all ensemble members of the 10-year long hindcast/prediction that are archived at





http://cmip-pcmdi.llnl.gov/cmip5/data_portal.html are used (see Table 1 for more information).
Each ensemble member was generated by slightly different initial conditions. Here we focus on
September Arctic (seasonal minimum) and Antarctic (seasonal maximum) sea ice. The reasons
that we focus on September Antarctic sea ice, rather than the month of seasonal minimum like
the Arctic are 1) sea ice in the Antarctic largely melts away (confined to the coastal Antarctica)
during the seasonal minimum (i.e. February or March), and 2) September sea ice extent has a
significant increasing trend.

Satellite-derived sea ice extent and concentration in the Arctic and Antarctic are used to

evaluate the CMIP5 decadal hindcast. They are obtained from the National Snow and Ice Data
Center, which are derived from the Nimbus-7 Scanning Mutichannel Microwave Radiometer
(SSMR), and DMSP Special Sensor Microwave/Imager (SSM/I), and Special Sensor Microwave
Imager and Sounder (SSMIS) sensors (Comiso 2000; Fetterer et al., 2002, 2010). Because the
observation and models have different horizontal resolution (see details in Table 1), before
performing the assessment we interpolate all the data (satellite observation and model
simulations) to horizontal resolution of 1 degree. The multi-model ensemble mean (MMEE) is
calculated based on the equally weighted average of 69 total ensemble members (Table 1).
**3. Prediction skill of CMIP5 decadal hindcasts**
**3.1 Arctic sea ice**

We evaluate the model simulation and prediction skill by comparing sea ice extent between

each individual model and satellite observations. Figure 2 shows the time series of September
Arctic sea ice extent from the simulation of the 10-year hindcast for each model and observation
from 1981 to 2015. It is immediately apparent that the models exhibit very different magnitudes
of September sea ice extent. CanCM4, CFSv2, GEOS-5 and GFDL-CM2.1 simulate a smaller





ice extent compared to the observation during the entire period; CFSv2 has the least sea ice cover
of any of the models. By contrast, BCC-CSM1.1, CCSM4, FGOALS-g2 and MIROC5 simulate
a larger ice extent. The simulated ice extent of HadCM3, IPSL-CM5A-LR and MPI-ESM-MR
are comparable to the observations, but they cannot reproduce the anomalously low sea ice cover
since 2007 (i.e., record lows in 2007 and 2012). We note that the models that are initialized with
values close to various estimates of sea ice state (direct and indirect full-field initialization, see
Table 1), drift towards their modeled sea ice state within a few year integrations, particularly
BCC-CSM1.1, CanCM4, CCSM4, CFSv2 and FGOALS-g2. Hence improved initializations do
not necessarily mitigate drift, although they significantly reduce the model bias at the initial step.
By contrast, the models that are initialized with various estimates of sea ice anomaly (direct and
indirect anomaly initialization) tend to have smaller drift problems during the integration.

To quantify the skill of each individual model and MMEE in predicting the evolution of sea

ice, we calculate the anomaly correlation coefficient (ACC) between the predicted and observed
September sea ice concentration anomaly in each grid box as follows.

$$ACC = \frac{\sum_{i=1}^{n}[P(i,t) - \overline{P}(t)] \cdot [O(i,t) - \overline{O}(t)]}{\sqrt{\sum_{i=1}^{n}[P(i,t) - \overline{P}(t)]^2 \cdot \sum_{i=1}^{n}[O(i,t) - \overline{O}(t)]^2}}$$

where P is the predicted sea ice concentration and $\overline{P}(t)$ is calculated as $\overline{P}(t) = \sum_{i=1}^{n} P(i,t)$; O is
the observed sea ice concentration and $\overline{O}(t)$ is calculated as $\overline{O}(t) = \sum_{i=1}^{n} O(i,t)$. i is the start
year and t is the lead year. Here the ACCs of the ensemble mean of each individual model and
MMEE for lead-times of 1, 3-5 and 6-8 years are discussed. For example, for the lead-time of
3-5 years, the data for the 1981 initialization is the average value of 1983-1985, the data for the
1986 initialization is the average value of 1988-1990, and so on. This means the adjacent data



points in the time-series have a time interval of 5-years, and this time-series is compared to the
average of the same three years in the observations.
For the lead-time of 1-year, for some models only scattered predictive skill (> 95%
significance) in forecasting September sea ice concentration anomalies are found, generally in
the arc around the periphery of the Arctic Basin extending from north of Alaska to northeast of
Siberia (top panel of Fig. 3). The MMEE shows small clustered areas of significant ACCs
between the Beaufort and eastern Siberian Seas, whereas areas near the central Arctic Ocean has
the least predictive skill (negative ACCs, Fig. 3l in the top panel). In general, the areas of
significant ACCs in CCSM4, MIROC5 and MPI-ESM-MR are similar to that of the MMEE.
For the lead-time of 3-5 years, the areas of significant predictive skill become broader for
the majority of the models compared to those of 1-year, covering large parts of the northern
Beaufort, Chukchi, eastern Siberian and Laptev Seas (bottom panel of Fig. 3). The exceptions are
CFSv2 and GEOS-5. CFSv2 has too little sea ice cover in the Arctic Ocean due to the
aforementioned drift problem. The ACCs of GEOS-5 for the lead-time of 3-5 year are even
smaller than those of 1-year for the area of ACCs exceeding the 95% confidence level. The
MMEE shows large clustered areas of significant ACCs in the arc around the Arctic Basin
extending from north of Alaska to north of Siberia (Fig. 3l in the bottom panel). Again, the
central Arctic Ocean towards the Canadian Archipelago and northern Greenland Sea shows the
least predictive skill.
The results for the lead-time of 6-8 years are broadly similar to those of the lead-time of 3-5
years, although the areas of significant predictive skill are relatively broader for the majority of
the models (not shown). The MMEE also shows enlarged areas of significant ACCs relative to



those of 3-5 year, i.e., along the eastern coast of the Greenland (not shown). Overall, the MMEE
has better prediction skill relative to individual models for all lead times, although the MMEE
does not universally outperform every single constituent models.

Figure 4 shows the predicted trend (slope of a linear regression) as a function of lead times

after applying a 3-year average to filter out high frequency variability. For each individual model,
the trend is calculated based on its ensemble mean (see No. of ensemble members in Table 1).
All the models reproduce the observed negative trend, except that BCC-CSM1.1 has positive
trend at the lead-time of 1-3 and 2-4 years. However, the simulated negative trends show very
different magnitude, ranging from about -0.2 to $-0.9 \times 10^6$ km$^2$ per decade. Compared to the
observation, there is a systematic underestimation of the decreasing trend throughout the
integration period for all decadal hindcasts. This is particularly true for the lead-time of 6-8 and
7-9 years (i.e., about $-0.6 \times 10^6$ km$^2$ per decade for the MMEE vs. $-1.2 \times 10^6$ km$^2$ per decade for the
observation), because those longer lead times are weighted towards inclusion of more recent
years in the observations with accelerated decline of Arctic sea ice.

To figure out to what extent the identified areas with significant ACCs at different lead times

are caused by the decadal decreasing trend, we remove the linear trend in the predicted and
observed sea ice concentration in each grid box. As shown in Fig. 5, after the trend is removed,
the areas with significant ACCs become much smaller relative to those of Fig. 3, especially for
the lead-time of 3-5 and 6-8 years. This suggests that high predictability found in Fig. 3 at longer
time scales is largely due to the decreasing Arctic sea ice in recent decades. Thus the relatively
long prediction skill over the areas of the northern Beaufort, Chukchi, eastern Siberian and
Laptev Seas is influenced by long-term sea ice reduction.





To further examine the prediction skill of Arctic sea ice variability in the context of regional
climate variability, we generate three sea ice extent indices: 1) the entire Arctic, 2) the north
Pacific, and 3) the north Atlantic. Sea ice variability in the north Pacific and north Atlantic is
modulated by different dominant decadal oscillations. Previous studies suggested that sea ice in
the Bering and Beaufort Seas is correlated with the Pacific Decadal Oscillation (PDO), which
has undergone a transition from a dominantly positive phase to a more negative phase in recent
decades (Lindsay and Zhang 2005; Zhang et al., 2010; Wendler et al., 2014). Sea ice in the north
Atlantic, particularly the ice export through Fram Strait and import from the Barents Sea, is
significantly affected by the phases of the North Atlantic Oscillation (e.g., Kwok, 2000; Rigor
and Wallace, 2004). Enfield et al. (2001) linked North Atlantic sea ice variability to the Atlantic
Multidecadal Oscillation (AMO) using the time frequency analysis of historical and paleo
records. Day et al. (2012) suggested that up to 30% of the north Atlantic sea ice decline during
1979-2010 might be attributed to the natural cycle of the AMO by analyzing five CMIP3 models.
Here we define the north Pacific sea ice index as the total September sea ice extent in the
Chukchi, East Siberian, and Laptev Seas (120°E-150°W and 62.5°N-80°N). The north Atlantic
sea ice index is defined as the total September sea ice extent in the Greenland, Norwegian, and
Barents Seas (40°W-80°E and 60°N-84°N, see boxes in Fig. 1). A 3-year average is also applied
to these indices.
The predictive skill for these indices is also measured by the anomaly correlation coefficient
between the model hindcast and observation. Figure 6 shows the ACC as a function of lead times
for the ensemble mean of each individual model and MMEE. To provide additional perspective
on the relative skill of the decadal experiments, the anomaly correlation coefficient of the
persistence prediction is also shown. Persistence prediction is the simplest way to produce a





forecast, which assumes sea ice state at the time of the forecast will not change. The horizontal
lines in Fig. 6 represent different confidence level. For the entire Arctic (Fig. 6a), the anomaly
correlation coefficient of most models exhibits certain predictive skill (> 95% significance),
except BCC-CSM1.1 for the lead-time of 1-3 and 2-4 years. Four models (CCSM4, FGOALS-g2,
GFDL-CM2.1 and MIROC5) show comparable or better predictive skill relative to the
persistence prediction for all the analyzed lead-times. The MMEE has more skillful results than
most of the individual model predictions during the entire period. The north Pacific sea ice index
has lower prediction skill and larger inter-model spread compared to those of the entire Arctic
index (Fig. 6c). In the north Pacific, only two models (GFDL-CM2.1 and MIROC5) show
comparable skill to the persistence prediction for the lead-time of 1-3 and 2-4 years. After 3-5
years, six models (CanCM4, CCSM4, FGOALS-g2, GFDL-CM2.1, MIROC5 and
MPI-ESM-MR) have better skill than the persistence prediction, which is also the case for the
MMEE. In general, the predictive skill of the north Atlantic sea ice index is poor compared to
both the entire Arctic and north Pacific indices, particularly for the lead-time from 3-5 to 5-7
years (insignificant ACCs). However, we note that in the north Atlantic sector all the models
show better predictive skill than the persistence prediction for the first three lead-times.
Additionally, all the models, except CanCM4, appear to have a re-emerging predictive skill for
the north Atlantic sea ice after 6-8 years (Fig. 6e). Overall, the MMEE has more skillful results
than that of the persistence prediction.

After removing the linear trend (Fig. 6b, d, f), the predictive skill of the above indices

decreases dramatically with very large inter-model spread. The MMEE only shows more skillful
results than the persistence prediction between 3-5 and 5-7 years for the north Pacific index.
**3.2 Antarctic sea ice**





Here we apply the same analysis in section 3.1 for Antarctic sea ice. Figure 7 shows time
series of September sea ice extent from the 10-year hindcast for each individual model and the
observations during 1981-2015. FGOALS-g2, GEOS-5, and MIROC5 produce significantly less
sea ice compared to the observation for the entire period with GEOS-5 having the smallest sea
ice extent of all the models. BCC-CSM1.1, CanCM4, and HadCM3 produce more sea ice
relative to the observations. The sea ice extent simulated by CCSM4, CFSv2, GFDL-CM2.1,
IPSL-CM5A-LR and MPI-ESM-MR is comparable to the observations, but they cannot
reproduce the gradual increase of Antarctic sea ice in recent years (e.g., Comiso et al., 2011). As
in the Arctic, the models that use direct and indirect full-field initialization tend to drift towards
their modeled sea ice state within a few years of initialization.
Figure 8 shows the anomaly correlation coefficient of each individual model and MMEE for
the lead-time of 1 and 3-5 years. For the 1-year lead-time, small scattered areas with predictive
skill greater than 95% confidence level in the Southern Ocean are found in most models. The
location of these scattered areas differs by model, although the MMEE shows small clustered
areas of significant ACCs in the central Weddell Sea (top panel of Fig. 8l). There is no
improvement for the predictive skill for most models and the MMEE as the lead-time increases
to 3-5 years (bottom panel of Fig. 8) and 6-8 years (not shown). Overall, the predictive skill of
the MMEE does not outperform most models for all the lead-times.
The observed and predicted trends for different lead times are shown in Fig. 9. The observed
trends are positive for all the lead-times, and increase to ~$0.35 \times 10^6$ km$^2$ per decade as recent
years are considered. By contrast, most models show negative trends, i.e., BCC-CSM1.1 has
negative trends ranging from -$0.6 \times 10^6$ km$^2$ to -$1 \times 10^6$ km$^2$ per decade. CCSM4 and FGOALS-g2
have increasing trends before 3-5 year and 5-7 year leads, respectively, but decreasing trends



thereafter. CFSv2 shows increasing trends after 2-4 year leads. However, these three
positive-trending models cannot simulate the magnitude of observed positive trends.
Again, we remove linear trends in both the model hindcast and observation, and then
calculate the ACC. After the linear trend is removed, the areas having significant predictive skill
become broader for the majority of the models compared to those of the raw data (Fig. 10 vs. Fig.
8), particularly for the lead-time of 3-5 and 6-8 years. Moreover, most models and the MMEE
have good predictive skill in the Ross Sea. As indicated by the MMEE, much of Antarctica's
coast has poor predictive skill (negative ACCs, Fig. 8l).
Here we generate three regional sea ice extent indices: 1) the entire Antarctic, 2) the
central-eastern south Pacific and 3) the south Atlantic. We define the central-eastern south
Pacific index as the total September sea ice extent in the eastern Ross, Bellingshausen and
Amundsen Seas (165°W-75°W and 50°S-80°S) and the south Atlantic index as the total
September sea ice extent in the Weddell Sea (60°W-0° and 50°S-75°S, see boxes in Fig. 1).
Figure 11 shows the anomaly correlation coefficient as a function of lead times for the
ensemble mean of each individual model, the MMEE and the persistence prediction. For the
entire Antarctic, none of models can predict the observed sea ice variability (i.e., their
simulations are negatively correlated with the observations), except for CCSM4 and
GFDL-CM2.1, which show significant prediction skill (> 95% significance) at the lead-time of
1-3 years (Fig. 11a). Moreover, the persistence prediction is superior to the prediction of each
individual model and the MMEE. For the central-eastern south Pacific index, almost all the
models show poor predictive skill for almost all the lead-times, although CFSv2, GFLD-CM2.1
and HadCM3 exhibit significant skill at 1-3, 2-4 and 4-6 years, respectively. Unlike the entire



Antarctic, the MMEE of the central-eastern south Pacific shows better skill than that of the
persistence prediction, although neither is statistically significant (Fig. 11c). For the south
Atlantic index (Fig. 11e), almost all the models also do not have predictive skill (the ACCs are
not statistically significant), although CCSM4 has significant skill at the lead-time of 5-7 years.
However, the MMEE shows surprisingly significant skill, much better than the persistence
prediction, at 6-8 years (> 95% significance).

After removing linear trends in Fig. 11a, c, e, we note that there is no obvious improvement

in predictive skill for the entire Antarctic and the central-eastern south Pacific indice, but the
inter-model spread is increased (Fig. 11b, d). It is also noted that for the south Atlantic index, the
MMEE shows significant skill after 4-6 years (Fig. 11f).
**4. Discussion and conclusion**

This assessment provides a snapshot of the interannual to decadal predictability of sea ice in

the Arctic and Antarctic for the current-day coupled global climate models as part of the CMIP5
decadal prediction experiment.

Our evaluation shows that for many models, there are substantial discrepancies between the

decadal hindcast and observed September sea ice extent. For instance, in the Arctic, as
mentioned previously, CFSv2 dramatically underestimates September sea ice cover, leading to
pronounced drift in the first three years of the decadal hindcast. In contrast, CFSv2 simulates a
larger March sea ice extent ($2\text{-}3\times10^6$ km$^2$ more than the observation, not shown). Hence there is
an excessive melt of sea ice through the melting season which is due to not only the
underestimate of observed September sea ice cover, but also the overestimate of observed March
sea ice cover (March minus September). Such large errors have the potential to propagate





through other components of the climate system. This excessive melt greatly increases
freshwater in the Arctic Ocean and export of fresh water through Fram Strait into the northern
Atlantic. Following Koenigk et al. (2007), we calculate the freshwater export through Fram strait
using the following formula:

$$Q = \int_{z=B}^{T} \int_{x=x0}^{x1} u \left( \frac{S_{ref} - S}{S_{ref}} \right) dx dz$$

where B is the bottom of the ocean layer (here B = 100m), T is ocean surface; x0 and x1 are end
points of the selected cross-section (here the cross-section is along 74°N and between 30°W and
10°E); S, Sref are salinity and reference salinity (Sref = 34.8 psu). As shown in Fig. 12, there is a
pronounced increase of the freshwater export through Fram Strait into the northern Atlantic
during the first 4 years of integration, although the amount of the freshwater export decreases
gradually after that. Such freshwater propagation into the North Atlantic results in a weakening
of deep water formation in the Greenland Sea. Also shown in Fig. 12, the volume transport of the
Atlantic Meridional Overturning Circulation (AMOC) at 40°N in CFSv2 (which is too weak at
the beginning of the integration) decreases substantially during the decadal hindcast (4Sv after
10-year integration), which is a factor of 3-4 smaller than the observation (18.7Sv in
Cunningham et al., 2007; 17.2Sv in Smeed et al., 2014; McCarthy et al., 2015). Thus incorrect
prediction of sea ice in the Arctic could influence the AMOC prediction, which is a key source of
decadal predictability for European climate (Jackon et al., 2015), and has global impacts at
longer timescales.
It is well-known that brine rejection during sea ice growth strongly influences the formation
of the Antarctic Bottom Water (AASW). In the Antarctic, as mentioned previously, GEOS-5
simulates much less September sea ice extent, a factor of about 6 less than the observation,



which is also the case for March sea ice extent (not shown). The underestimation of sea ice
coverage might result in insufficient brine rejection through the freeze-up period in the GEOS-5.
This insufficient brine rejection is due to not only the underestimate of observed September sea
ice cover alone, but also the underestimate of observed March sea ice cover. Export of AABW
constitutes a key component of the meridional overturning circulation in the Southern Ocean
(Lumpkin and Speer 2007). The systematic underestimation of sea ice coverage results in a
weaker Deacon Cell in the Southern Ocean (~4Sv, Fig. 13) compared to the estimate of 20Sv
from Döös et al. (2007). Therefore, models that have large biases in simulating sea ice extent
(e.g., CFSv2 for the Arctic, GEOS-5 for the Antarctic) result in degraded predictive skill in sea
ice as well as other variables.
By performing the anomaly correlation analysis, we found that in the Arctic most models
only show small clustered areas with significant predictive skill at the lead-time of 1-year. As the
lead-time increases, for most models, the areas with significant predictive skill expand, covering
much of the northern Beaufort, Chukchi, eastern Siberian, and Laptev Seas. Such expansion is
largely due to the fact that almost all the models can predict observed negative trends of Arctic
sea ice, although the magnitude of the trend simulated by most models is still smaller than
observed. After the linear trend is removed, the areas with significant predictive skill at longer
time scales shrink greatly.
The analysis of regional indices suggests that sea ice in the Atlantic side has lower
predictability than that of the Pacific side. This is perhaps counterintuitive, since the AMO is
well predicted compared to the PDO (Kim et al., 2012). We do note that, for the Atlantic side of
the Arctic, most models show re-emerging predictive skill at the lead-time of 6-8 years. This
might be associated with the existence of interior AMOC pathways. A stronger (weaker) AMOC



results in warming (cooling) in the subpolar gyre after several years, contributing to enhanced
decadal predictability of sea ice in the north Atlantic sector (e.g., Mahajan et al. 2011; Zhang and
Zhang, 2015). In contrast to our results focusing on September sea ice, some idealized modeling
studies (Koenigk and Mikolakewicz 2009; Koenigk et al., 2012), which assess predictive skills
relative to their model climate, suggested annual and decadal mean sea ice concentration has
higher potential predictability for the Atlantic side than that of the Pacific side. Germe et al.
(2014) showed that the potential predictability of the winter Arctic sea ice extent comes mainly
from the Atlantic sector, while the Pacific sector seems unpredictable beyond the first year.
Further research is needed to explore the differences across model configurations.
By contrast, Antarctic sea ice does not show promising predictive skills at longer time scales.
Unlike in the Arctic counterpart, there is no obvious change in the areas showing significant
predictive skill as the lead-time increases. This might be because most models cannot predict
observed increasing Antarctic sea ice in recent decades. Instead almost all decadal hindcasts
predict a decrease of Antarctic sea ice, which is also true for the simulation in recent decades and
in response to forced simulations that include increased greenhouse gases in the atmosphere (e.g.,
Liu and Curry, 2010; Turner et al., 2013; Shu et al., 2015). Further investigating a range of other
variables such as simulated sea ice thickness, sea ice velocity, near surface wind, and ocean
stratification will help elucidate the reasons why the trends in these models are different from
observations. However, after the trend is removed, most models suggest that large parts of the
eastern south Pacific do have some predictive skill. Previous studies (e.g., Liu et al., 2002) have
showed that the intensification of the Hadley Circulation in the eastern equatorial Pacific during
El Nino leads to an equatorward shift of the storm track in the eastern south Pacific. This leads to
the changes of the regional Ferrel Circulation in the eastern Pacific, which cause an anomalous



poleward mean meridional heat flux into the sea ice zone in the eastern south Pacific and limits
sea ice growth there. Thus, relatively good sea ice predictability in the eastern south Pacific
might be related to the ENSO teleconnection. Holland and Raphael (2006) further showed that a
number of climate models have the ability in simulating the observed ENSO teleconnection in
sea ice in the eastern south Pacific and Atlantic. The analysis of regional indices suggests that the
MMEE has skillful results in the south Atlantic beyond 4-6 years, whether or not the trend is
removed.
An issue in this assessment is the relatively small sample size because of the limited number
of start years of the decadal prediction experiment. To promote both the science and practice of
decadal prediction, the CMIP phase 6 recommends ensembles of 10-year hindcast/prediction for
all years from 1960 to the end of the CMIP6 period (10 members recommended), which will be
helpful to obtain better statistics. As demonstrated in this study and previous studies, large biases
in models strongly influence sea ice prediction at decadal time scales. Thus continued efforts are
needed to identify, understand and reduce model errors, i.e., Kharin et al. (2012) demonstrated a
technique to correct non-linear drifts in decadal hindcasts. Some multi-model studies put efforts
on this issue for some climate variables (e.g., Bellucci et al., 2014; Doblas-Reyes et al., 2013;
Goddard et al., 2012).
Recent studies suggested that different initialization approaches and the density of
observations used in the initialization significantly affect the predictability of sea ice. Zunz et al.
(2015) tested three initialization approaches and found that the spread of ensembles at decadal
time scales can be reduced when more complicated data assimilation procedures and denser
observations are used to initialize the hindcasts. To date, only limited models have implemented
initialization of sea ice concentration (see Table 1 for details). Moreover, to better predict sea ice,



the accurate sea ice initialization requires not only sea ice concentration, but also variables (i.e.
sea ice thickness) influence surface energy fluxes and, thereby, ocean-atmosphere interaction. At
seasonal timescales, the initialization of sea ice thickness has been shown to be crucial for
summer prediction (e.g., Day et al. 2014b). Some studies (e.g., Blanchard-Wrigglesworth et al.
2011a; Koenigk and Mikolajewicz, 2009) suggested that the persistence of sea ice thickness
anomalies is much higher than that of sea ice concentration anomalies. Higher predictability of
Arctic sea ice thickness (volume) with respect to that of Arctic sea ice cover has been found at
longer time scales (e.g., Guemas et al., 2014). However, sea ice thickness has not yet been
initialized in CMIP5 models because of sparse observations. In this assessment, based on Table 1,
11 CMIP5 models can be separated into two groups: direct and indirect sea ice initialization. The
direct initialization includes CanCM4, CFSv2 and GEOS-5. Other models are indirect
initialization. Based on this division, we cannot conclude that the models initialized directly has
better performance on predictive skills compared to those initialized indirectly. CanCM4 has
broader area with significant predictive skill at longer lead-times (Figure 3). Its predictive skill is
better than some models (e.g., BCC-CSM1.1, HadCM3, IPSL-CM5A-LR), comparable with
CCSM4 and GFDL-CM2.1, but worse than MIROC5 and MMEE. On the other hand, CFSv2 has
strong model drift so that the predicted sea ice is substantially less than the observations.
GEOS-5 has nearly no skill on predicting observed sea ice variability. From this comparison, it is
not clear whether direct sea ice initialization is better than indirect sea ice initialization. Other
processes important for simulating sea ice evolution include the ocean below sea ice (i.e.,
temperature and salinity), which, due to its long persistence time, provides constraints on
predictions of sea ice at longer time scales. Thus, efforts should be devoted to further
development of initialization of the Arctic Ocean and Southern Ocean, which requires sufficient



observations and improved assimilation methods.
**Acknowledgements**
This research is supported by the NOAA Climate Program Office (NA14OAR4310216) and the
NSFC (41176169).

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

**Figure Captions:**
Figure 1. Linear trends of September sea ice concentration for (a) Arctic and (b) Antarctic during
the period of 1979-2014. The meshed areas denote the trends above 95% confidence level. Boxes
indicate the areas used to generate the regional sea ice indices.
Figure 2. Time series of September Arctic sea ice extent (seasonal minimum) from the
simulations of the 10-year hindcast for each ensemble member of each individual model (thin
gray line), the ensemble mean of each individual model (thick red line) and satellite observation
(black line) from 1981 to 2015.
Figure 3. Anomaly correlation coefficients between the simulated and observed Arctic
September sea ice concentration anomalies for the lead-time of 1-year (top panel) and 3-5 years
(bottom panel). The correlation coefficient 0.61, 0.73 and 0.88 represents 90%, 95% and 99%
confidence levels, respectively.
Figure 4. The predicted trends (slope of a linear regression) of September Arctic sea ice extent
anomalies as a function of the lead-time after applying a 3-year average.





Figure 5 same as Figure 3, but for detrended September sea ice concentration anomalies.
Figure 6. Anomaly correlation coefficients between the simulated and observed Arctic
September sea ice extent anomalies for the three regional indices (the entire Arctic, Pacific and
Atlantic) as a function of the lead-time. The top and bottom panels are the original and detrended
time series, respectively. The horizontal dashed and solid lines represent 90%, 95% and 99%
confidence levels, respectively. The thick gray line is the persistence prediction.
Figure 7. Time series of September Antarctic sea ice extent (seasonal minimum) from the
simulations of the 10-year hindcast for each ensemble member of each individual model (thin
gray line), the ensemble mean of each individual model (thick red line) and satellite observation
(black line) from 1981 to 2015.
Figure 8. Anomaly correlation coefficients between the simulated and observed Antarctic
September sea ice concentration anomalies for the lead-time of 1-year (top panel) and 3-5 years
(bottom panel). The correlation coefficient 0.61 ,0.73 and 0.88 represents 90%, 95% and 99%
confidence levels, respectively.
Figure 9. The predicted trends (slope of a linear regression) of September Antarctic sea ice extent
anomalies as a function of the lead-time after applying a 3-year average.
Figure 10. same as Figure 8, but for detrended September sea ice concentration anomalies.
Figure 11. Anomaly correlation coefficients between the simulated and observed Antarctic
September sea ice extent anomalies for the three regional indices (the entire Antarctic, eastern
Pacific and Atlantic) as a function of the lead-time. The top and bottom panels are the original
and detrended time series, respectively. The horizontal dashed and solid lines represent 90%, 95%
and 99% confidence levels, respectively. The thick gray line is the persistence prediction.

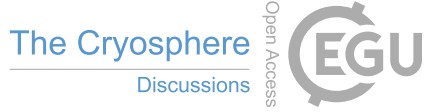

Figure 12. (a) Freshwater export through Fram Strait (the cross-section along 74°N and between
30°W and 10°E), (b) Atlantic Ocean meridional overturning streamfunction in September
averaged for all decadal hindcasts from 1981 to 2015 for the CFSv2 (upper panel) and (c) time
series of stream function averaged over 40-55N, 500-1500m as indicated by the black in upper
panel (lower panel). The thin gray line represents each ensemble member and the thick black line
represents the ensemble mean.
Figure 13. Atlantic Ocean meridional overturning streamfunction in September averaged for all
decadal hindcasts from 1981-2015 for GEOS-5 (upper panel) and time series of stream function
averaged over 45-70N, 500-2000m as indicated by the black in upper panel (lower panel). The
thin gray line represents each ensemble member and the thick black line represents the ensemble
mean.



Table 1 Summary of initialization methods and data sources used for the CIMP5 decadal
hindcast/prediction

| Model | Resolution (sea ice model) | Ensemble members | Sea ice assimilation method and data source |
|---|---|---|---|
| BCC-CM1.1 | 1 lon x 1-1/3 lat | 4 | None, but the initial sea ice indirectly influenced by nudging T to SODA ocean reanalysis |
| CanCM4 | ~2.8 lon x 2.8 lat | 10 | Full-field using SIC from HadISST1.1 and SIT from model-based climatology (Merryfield et al., 2013) |
| CCSM4 | 0.9 lon x 1.25 lat | 10 | Full-field using bias-corrected CORE2-forced ocean hindcast |
| CFSv2 | 0.5 lon x 0.5 lat | 4 | Full-field using NCEP climate forecast system reanalysis |
| FGOALS-g2 | 1 lon x 1 lat | 3 | None, but the initial sea ice indirectly influenced by nudging T and S to an ocean reanalysis |
| GEOS-5 | 1 lon x 1 lat | 3 | Full-field using GEOS-iODAS |
| GFDL-CM2.1 | ~1 lon x 0.75 lat | 10 | None, but the initial sea ice indirectly influenced by atmospheric and ocean data (Msadek et al. 2014) |
| HadCM3 | 1.25 lon x 1.25 lat | 10 | Anomaly-field using Met Office Hadley Centre sea ice data (HadISST) |
| IPSL-CM5A-LR | ~2 lon x 2 lat | 6 | None, but the initial sea ice indirectly influenced by the assimilation of T and S anomalies from observations |
| MIROC5 | 1 lon x 1 lat | 6 | None, but the initial sea ice is indirectly influenced by the assimilation of T and S from an objective analysis of Ishii and Kimoto (2009) |
| MPI-ESM-MR | ~0.4 lon x 0.4 lat | 3 | None, but the initial sea ice indirectly influenced by the assimilation of T and S anomalies from a forced ocean run using NCEP reanalysis (Müller et al., 2012) |

SIC: sea ice concentration; SIT: sea ice thickness, T: ocean temperature, S: salinity





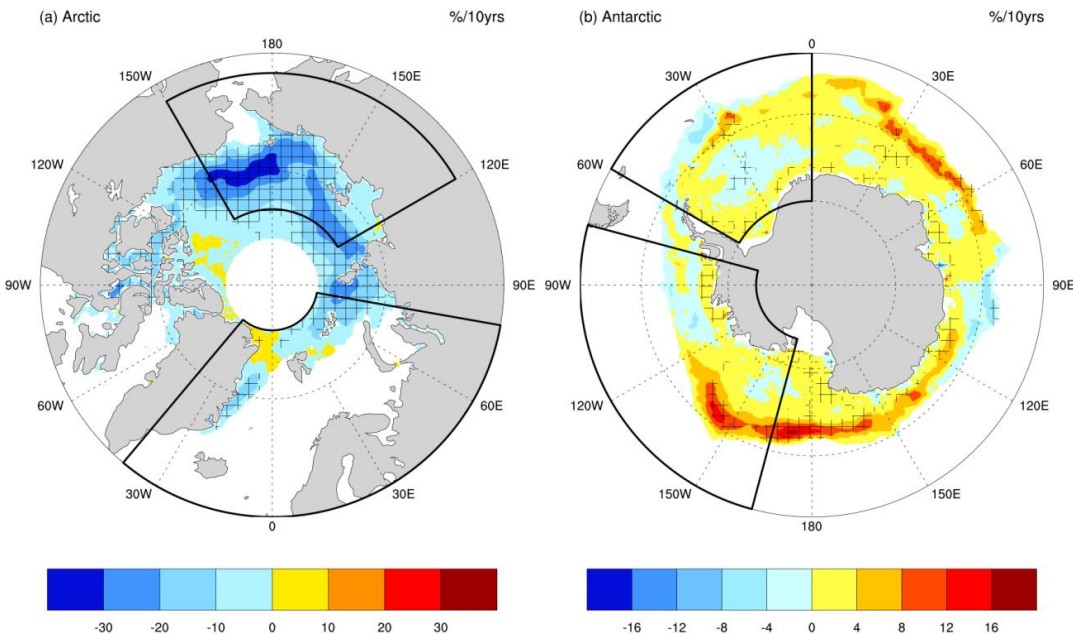

Figure 1. Linear trends of September sea ice concentration for (a) Arctic and (b) Antarctic during
the period of 1979-2014. The meshed areas denote the trends above 95% confidence level. Boxes
indicate the areas used to generate the regional sea ice indices.





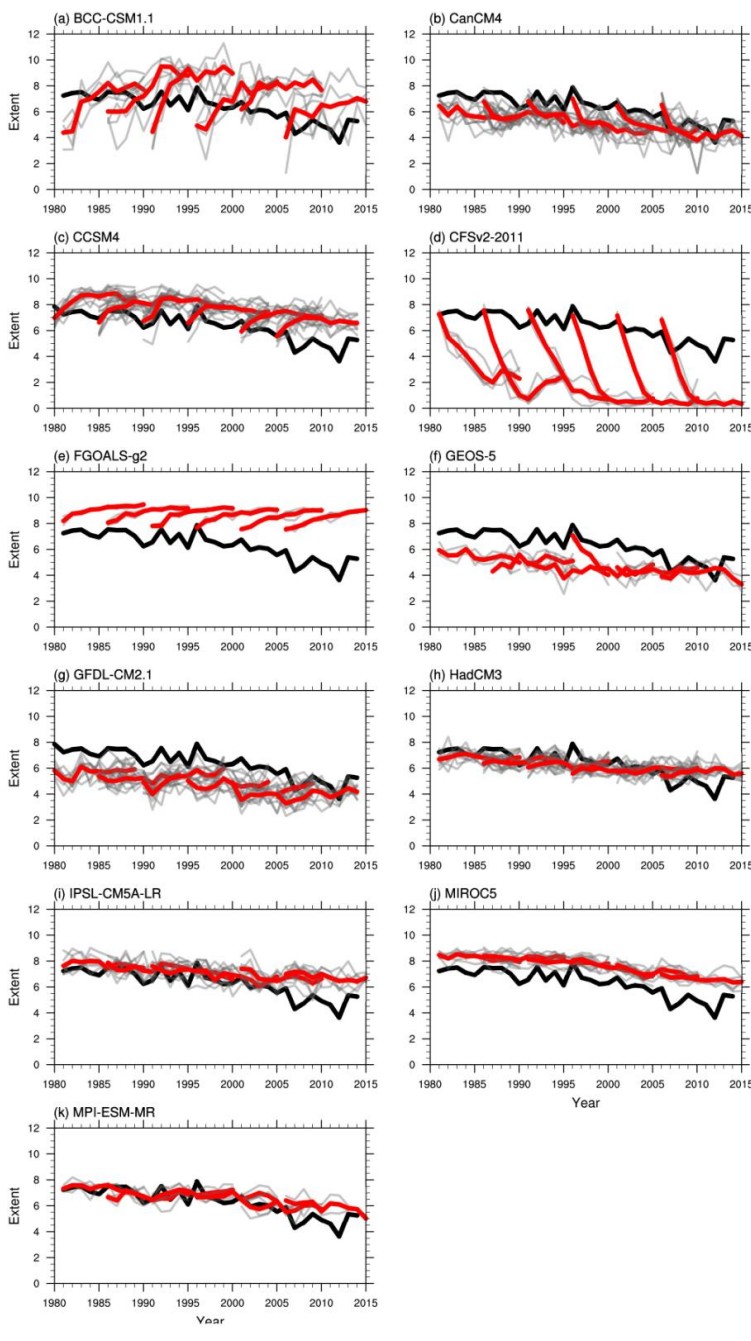


Figure 2. Time series of September Arctic sea ice extent (seasonal minimum) from the simulations of the 10-year hindcast for each ensemble member of each individual model (thin gray line), the ensemble mean of each individual model (thick red line) and satellite observation (black line) from 1981 to 2015.



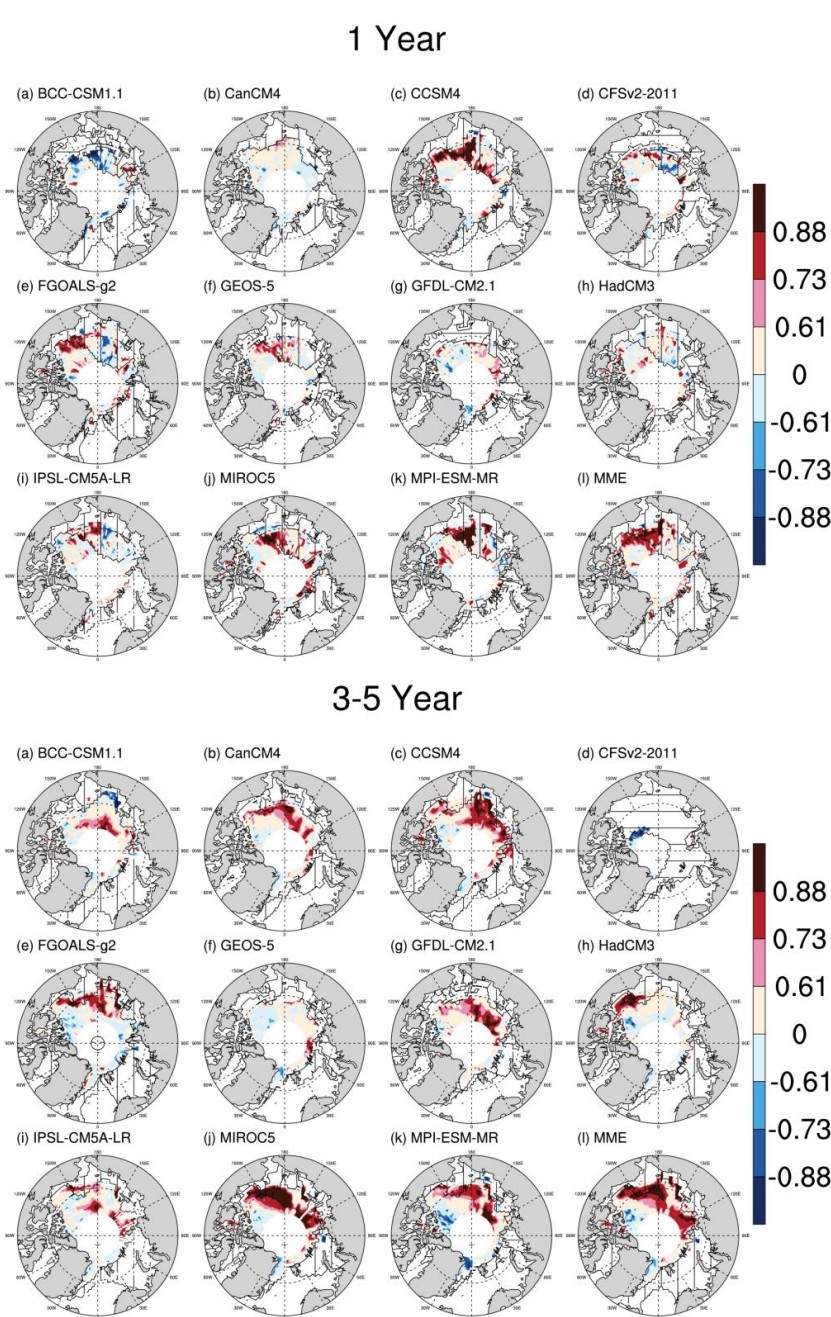

737

Figure 3. Anomaly correlation coefficients between the simulated and observed Arctic
September sea ice concentration anomalies for the lead-time of 1-year (top panel) and 3-5 years
(bottom panel). The correlation coefficient 0.61, 0.73 and 0.88 represents 90%, 95% and 99%
confidence levels, respectively.





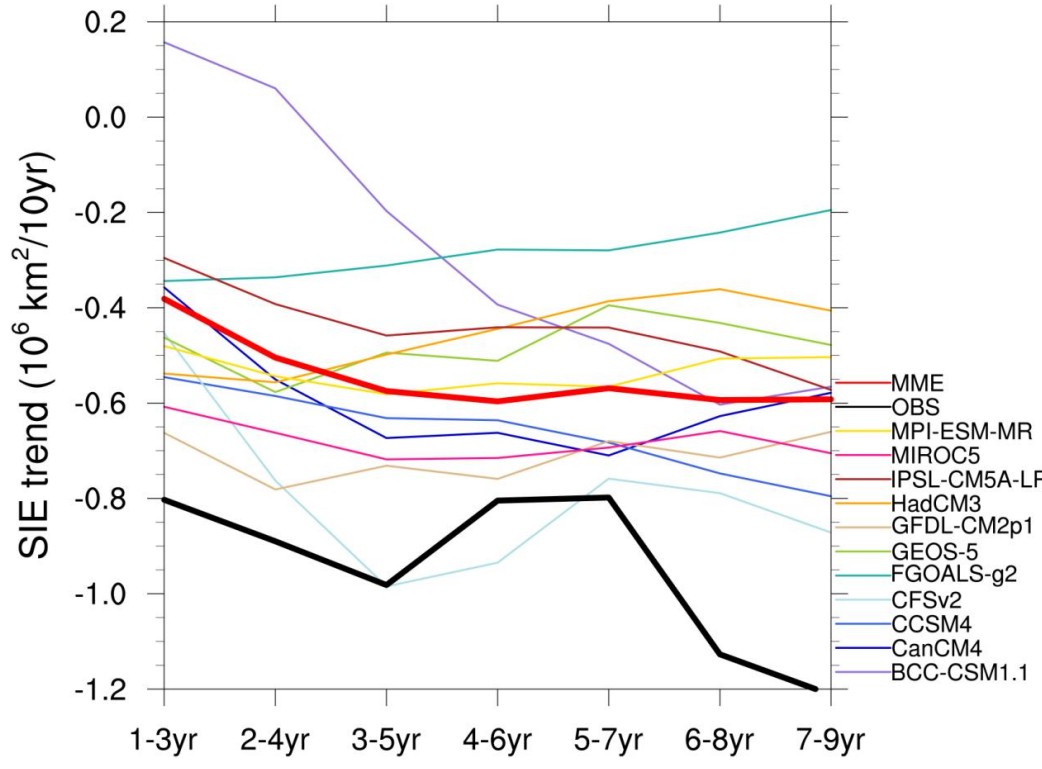


Figure 4. The predicted trends (slope of a linear regression) of September Arctic sea ice extent
anomalies as a function of the lead-time after applying a 3-year average.





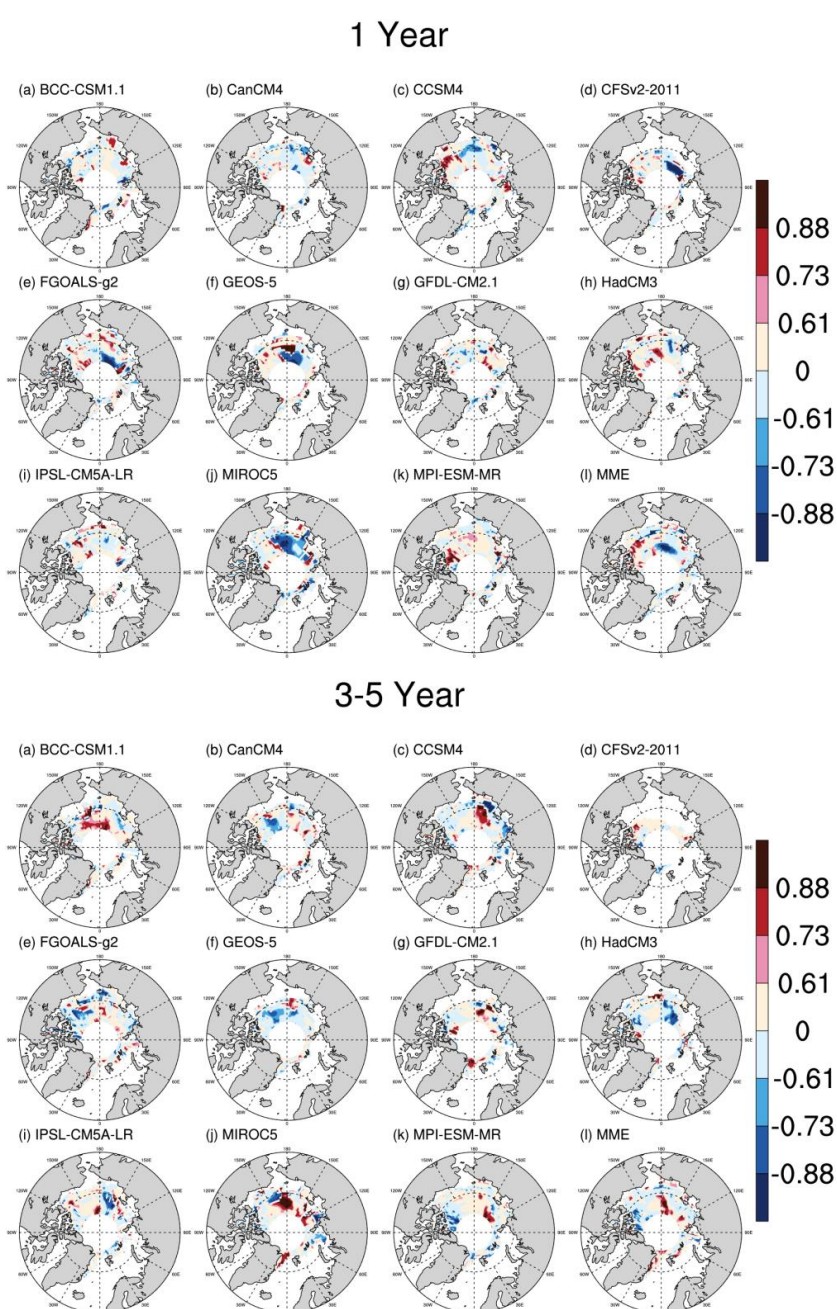


Figure 5 same as Figure 3, but for detrended September sea ice concentration anomalies.



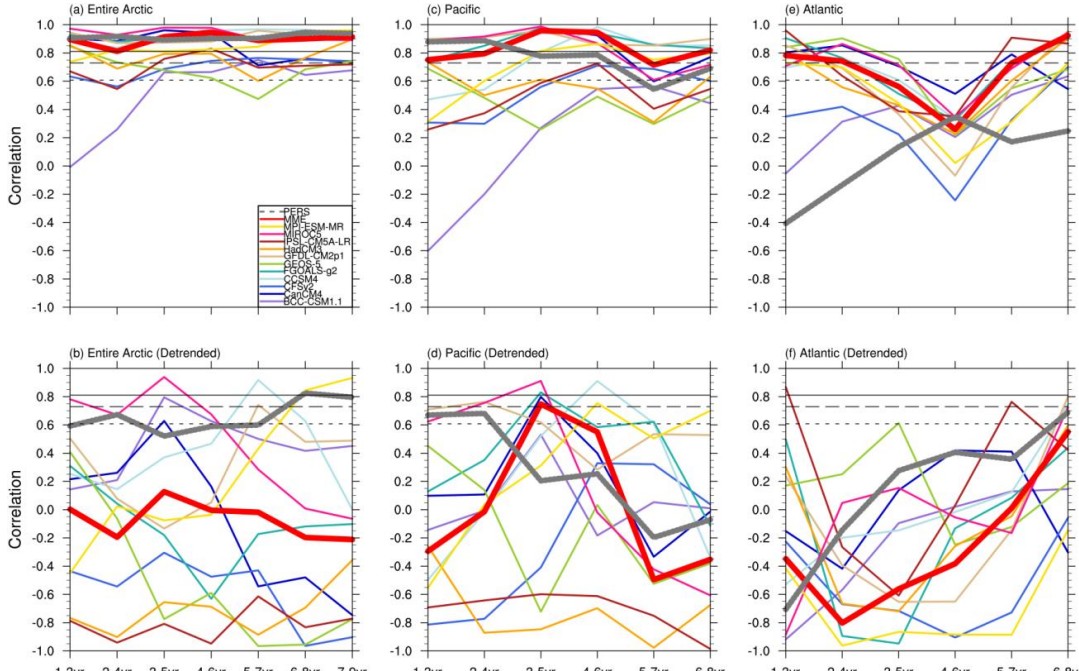


Figure 6. Anomaly correlation coefficients between the simulated and observed Arctic
September sea ice extent anomalies for the three regional indices (the entire Arctic, Pacific and
Atlantic) as a function of the lead-time. The top and bottom panels are the original and detrended
time series, respectively. The horizontal dashed and solid lines represent 90%, 95% and 99%
confidence levels, respectively. The thick gray line is the persistence prediction.






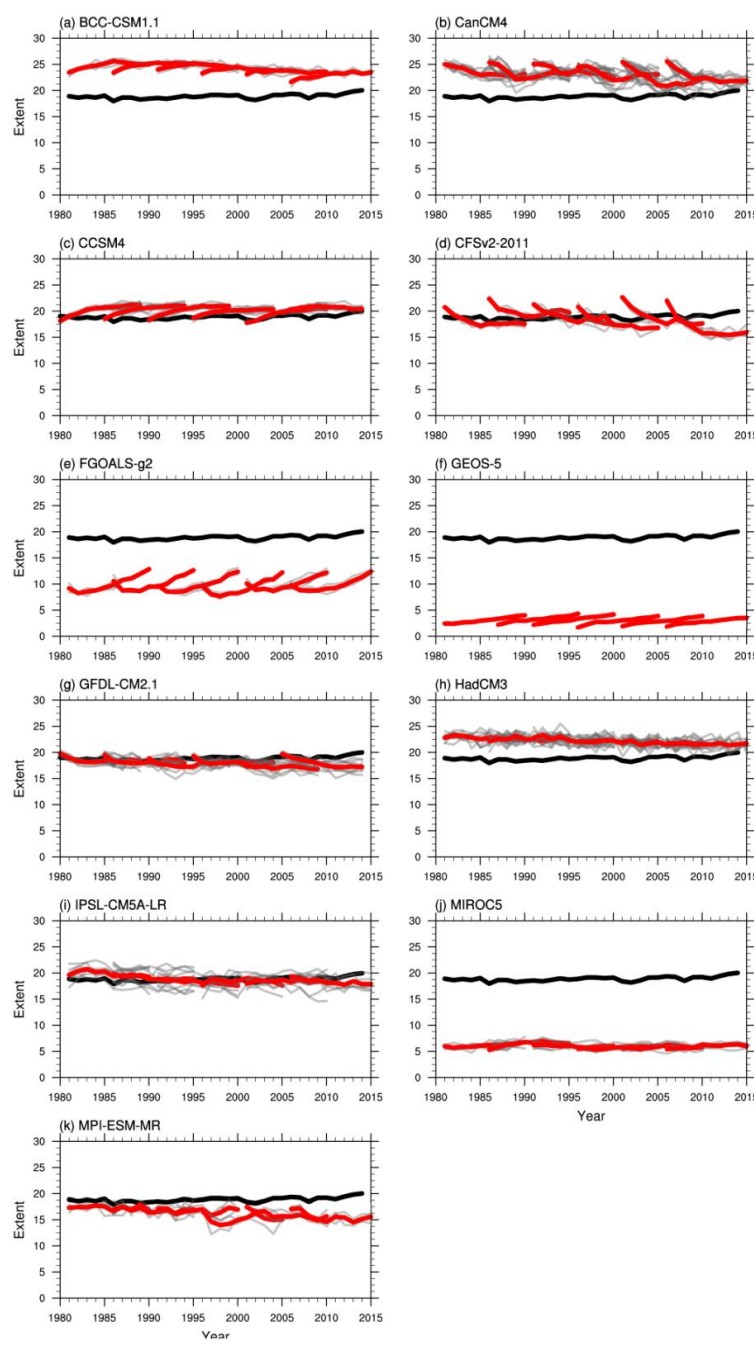

Figure 7. Time series of September Antarctic sea ice extent (seasonal minimum) from the
simulations of the 10-year hindcast for each ensemble member of each individual model (thin
gray line), the ensemble mean of each individual model (thick red line) and satellite observation
(black line) from 1981 to 2015.





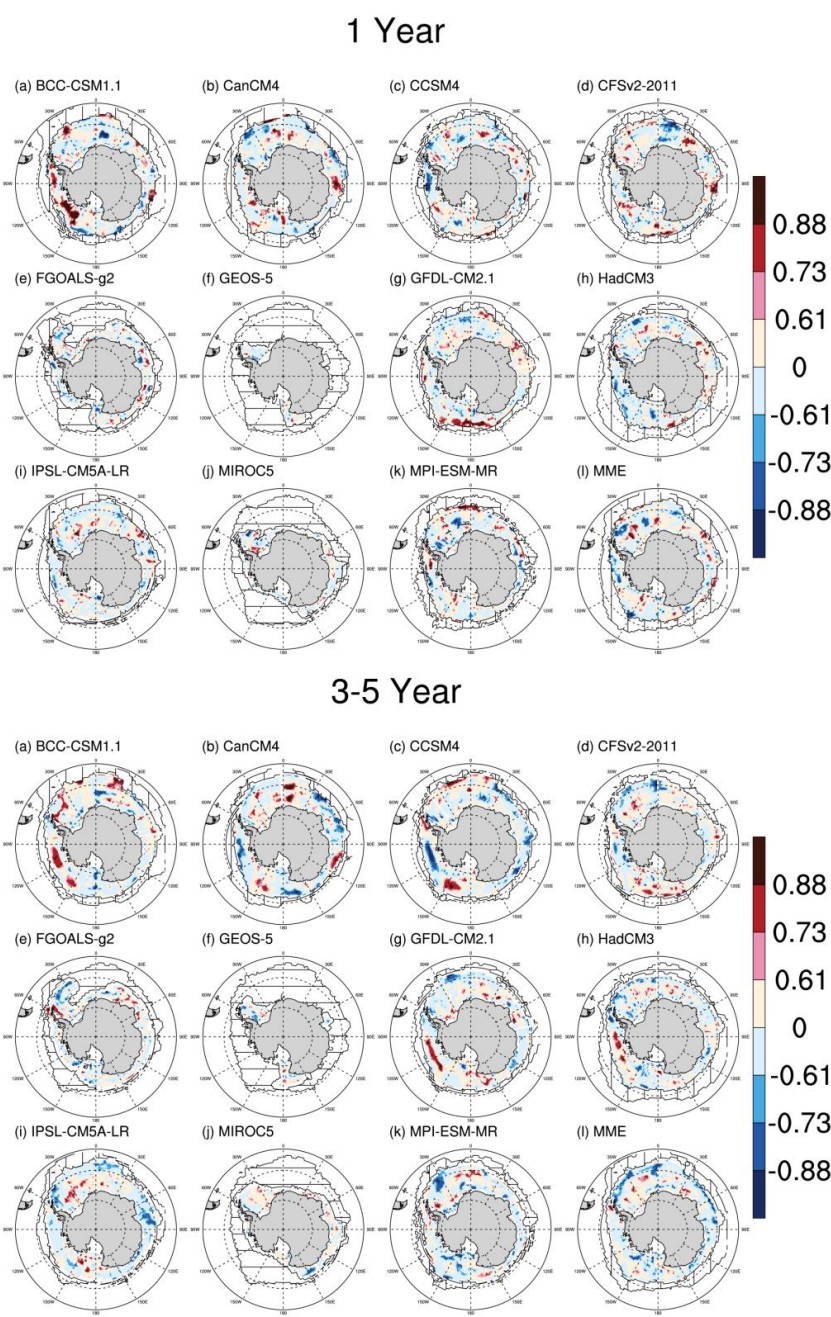

Figure 8. Anomaly correlation coefficients between the simulated and observed Antarctic
September sea ice concentration anomalies for the lead-time of 1-year (top panel) and 3-5 years
(bottom panel). The correlation coefficient 0.61 ,0.73 and 0.88 represents 90%, 95% and 99%
confidence levels, respectively.


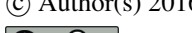

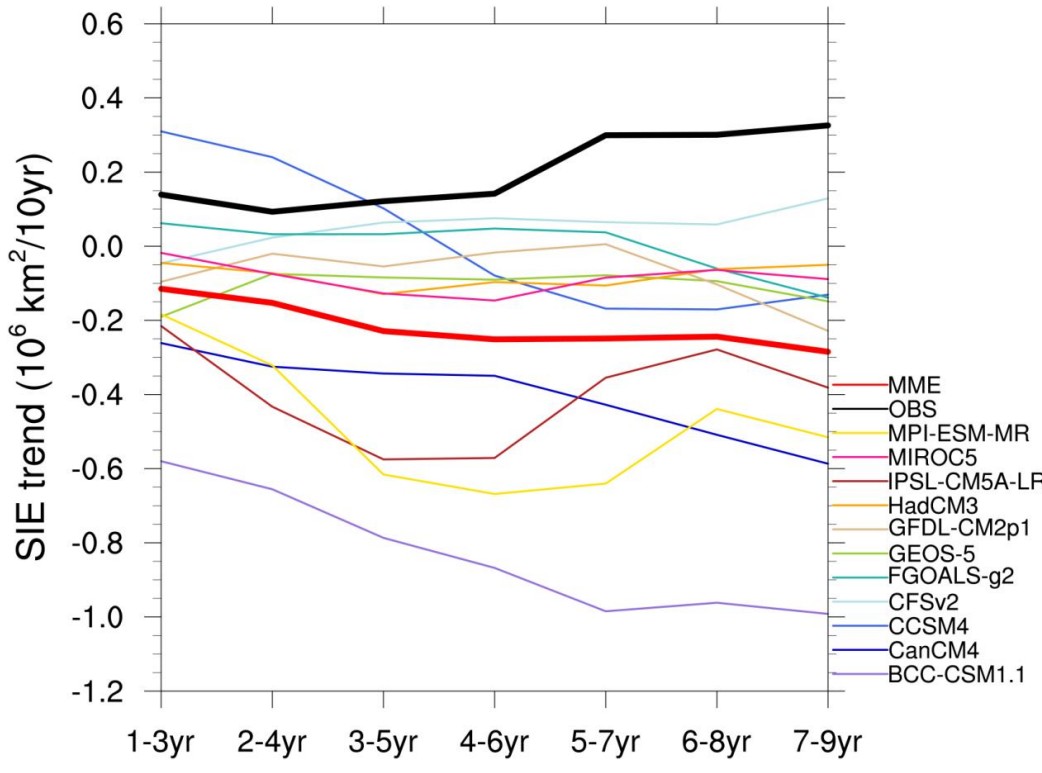

Figure 9. The predicted trends (slope of a linear regression) of September Antarctic sea ice extent
anomalies as a function of the lead-time after applying a 3-year average.



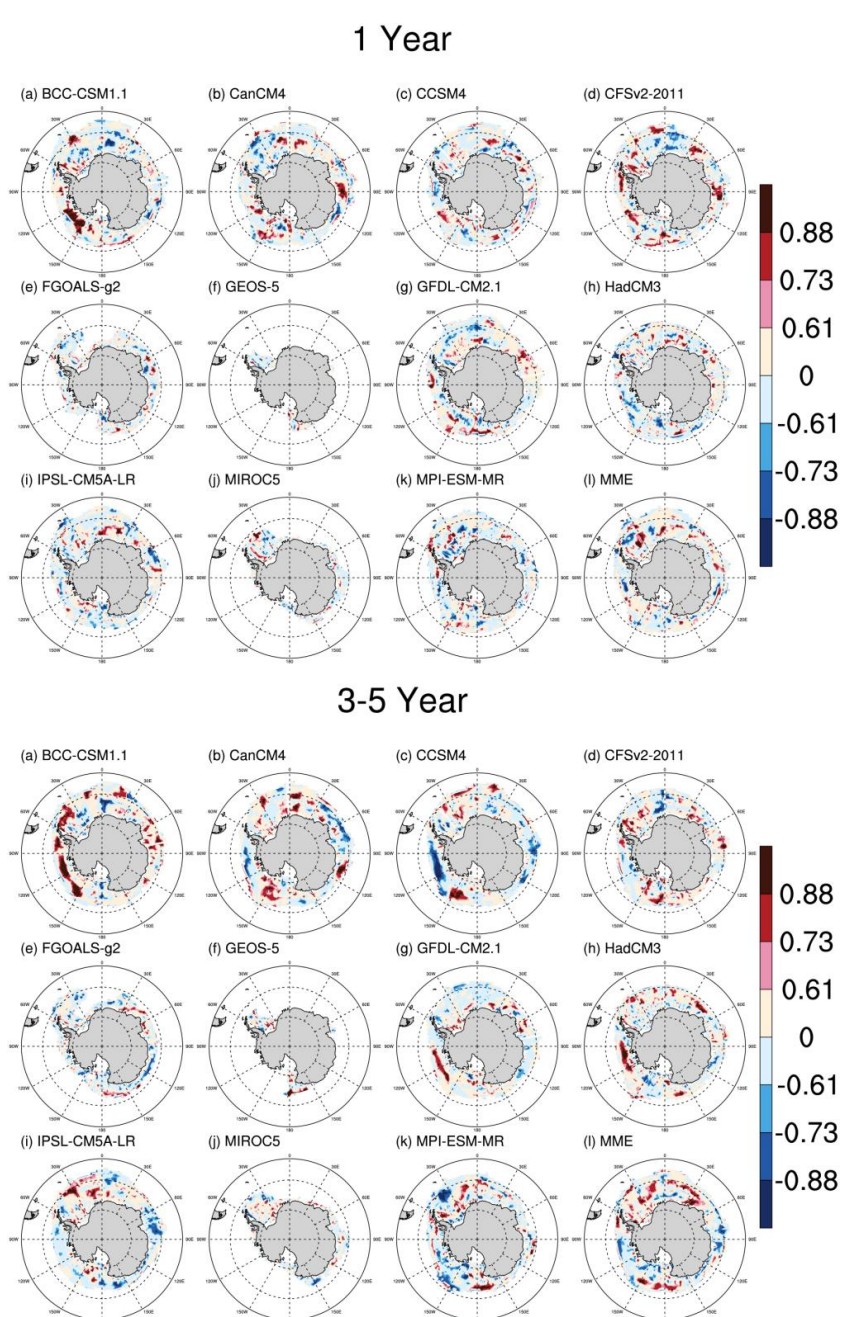

Figure 10. same as Figure 8, but for detrended September sea ice concentration anomalies.



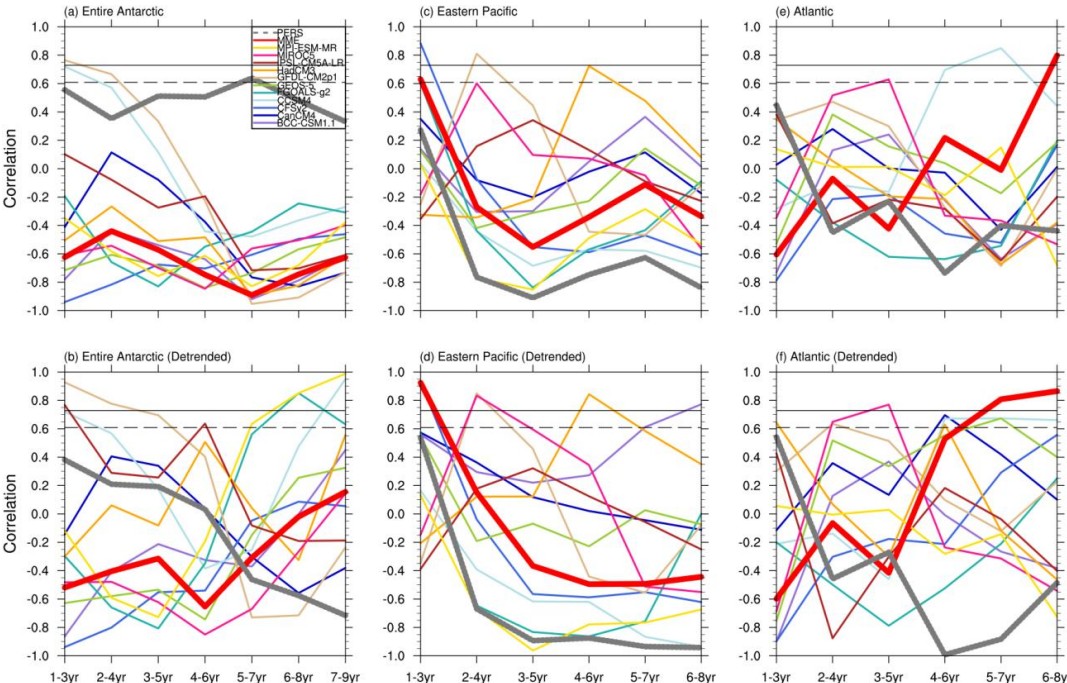

Figure 11. Anomaly correlation coefficients between the simulated and observed Antarctic
September sea ice extent anomalies forthe three regional indices (the entire Antarctic, eastern
Pacific and Atlantic) as a function of the lead-time. The top and bottom panels are the original
and detrended time series, respectively. The horizontal dashed and solid lines represent 90%,
95% and 99% confidence levels, respectively. The thick gray line is the persistence prediction.





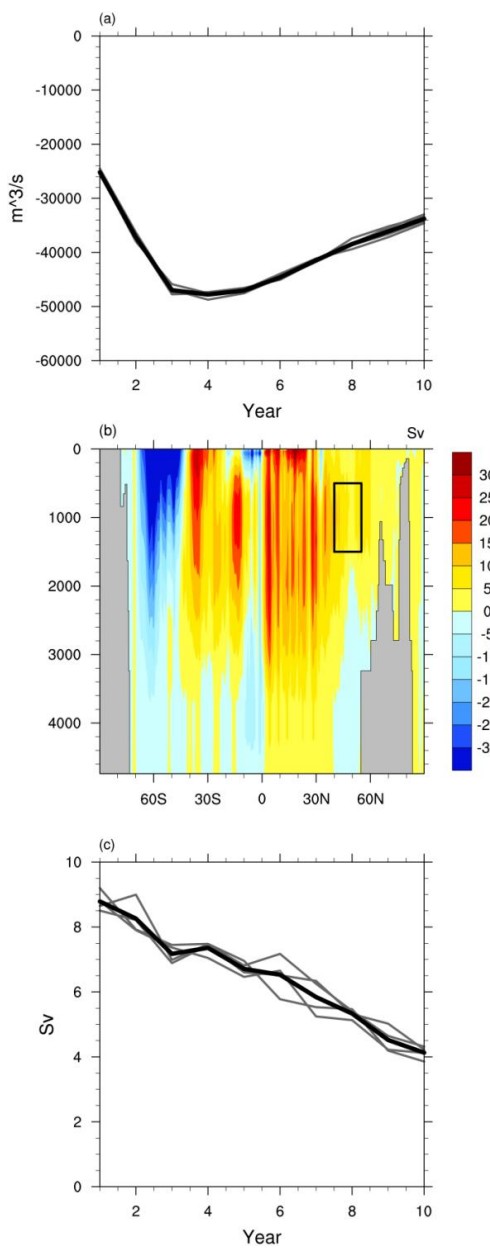

Figure 12. (a) Freshwater export through Fram Strait (the cross-section along 74°N and between 30°W and 10°E), (b) Atlantic Ocean meridional overturning streamfunction in September averaged for all decadal hindcasts from 1981 to 2015 for the CFSv2 (upper panel) and (c) time series of stream function averaged over 40-55N, 500-1500m as indicated by the black in upper panel (lower panel). The thin gray line represents each ensemble member and the thick black line represents the ensemble mean.





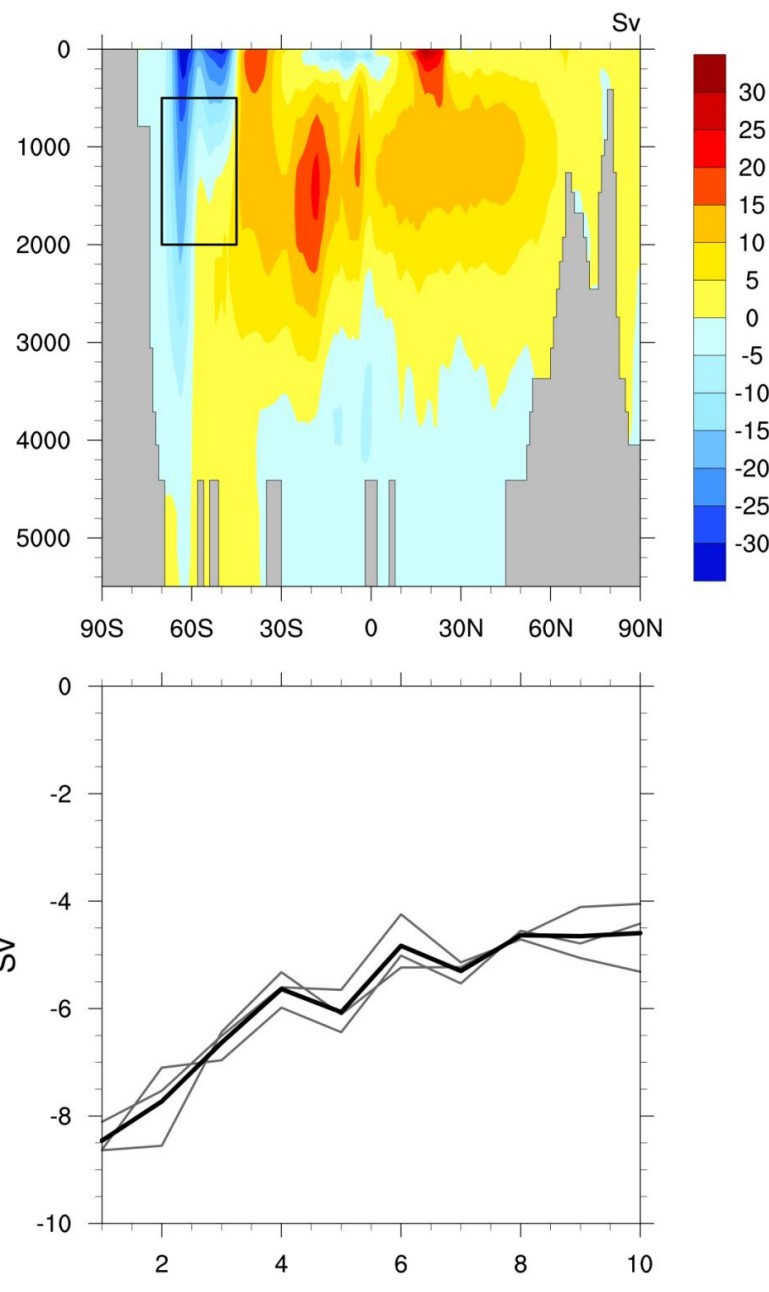

Figure 13. Atlantic Ocean meridional overturning streamfunction in September averaged for all decadal hindcasts from 1981 to 2015 for GEOS-5 (upper panel) and time series of stream function averaged over 45-70N, 500-2000m as indicated by the black in upper panel (lower panel). The thin gray line represents each ensemble member and the thick black line represents the ensemble mean.