# Peer review of "Assessment of Arctic and Antarctic Sea Ice Predictability in CMIP5 Decadal Hindcasts"

_The Cryosphere, 2016_

## Referee Comment (RC1) · Anonymous Referee #1 · 18 Jun 2016

The authors present an evaluation of sea ice covers in decadal hindcasts run with 11 models for CMIP5. They consider September Arctic and Antarctic sea ice concentration and extent, compares with observations over the period 1980-2006 (+10 years), and evaluate the ability of the models (and the multi-model ensemble mean) to capture the raw fields, the trends, regional features and anomalies relative to the trends.

Presenting an evaluation of the CMIP5 decadal experiments is a good thing per se, thus this paper is useful and timely. It actually reads well and presents interesting results. This draft shows that the predictive skill at decadal time scale is poor for both Arctic and Antarctic September sea ice extent. Yet, some skill arises in the Arctic due to the decreasing trend (anthropogenically forced) that models are able to reproduce. In the Antarctic, the lack of predictive skill is attributed to the models not being able to capture the recent increasing trend.

It would have been interesting to compare predictive skill for maximum sea ice covers in both hemispheres, ie September Antarctic sea ice and March Arctic sea ice. Germe et al (2014) wrote a very interesting study on the predictability of winter Arctic sea ice based on a CMIP5 hindcast (not used in the present draft), which gave some clues on possible processes to investigate. September Arctic sea ice extent is in the spotlight, but few skill should be expected from CMIP5 decadal hindcasts besides the trend, due to the date of initialization early in the year (around January 1 for most systems I guess) and the predictability barrier documented in Day et al (2014b).

I noticed a few major points that should be addressed before possible publication : some methodological issues, some unclear figures, and a poor discussion of the results. These points are marked with an asterisk (*), and can be considered as "major comments".

1.Introduction

L51: "ie" -> "eg"

L51-52: the sentence sounds weird. Does it mean that sea ice treatment is important to reduced the remaining "large uncertainties for decadal climate prediction"?

L111: "implemented an experiment" -> "implemented an experimental framework..."

L113: "validation" -> "evaluation"

2.Models and data

*How did the authors select those 11 models ? Why not using e.g. CNRM-CM5 (documented in Germe et al., 2014) or EC-Earth?

*In the first paragraph, the authors explain that there are two approaches for initialization, full-field and anomaly initialization. On L163, they introduce direct vs indirect initializations, which are not defined here.

*There is no information on the start date (ie: time of the year when the hindcast

are initialized). Are all models initialized in early January? Earlier? During January, the Antarctic sea ice cover is low : could it be a reason for the lack of skill (e.g. no persistence possible)?

L139-141: Let's be honest : the main reason why we don't focus on the Antarctic sea ice minimum is because we know the models poorly simulate this minimum. See for instance Turner et al., 2013.

3.Prediction skill of CMIP5 decadal hindcasts

*A discussion on the significance of the predicted trends is missing, as those trends seem to be calculated based on only 6 points (this is actually not well explained). . .

L163 : see above : what is "direct and indirect full initialization"?

L165 : what is "improved initialization"? Compared to what?

L172-174 : the formula is wrong. Pbar and Obar should be divided by n to denote an average.

L200-202 : the contradiction in this sentence strikes me.

L210/212 : "lead-time" or "lead time"? The authors should be consistent.

L212-213, "those longer lead times are weighted towards inclusion of more recent years in the observations with accelerated decline of Arctic sea ice". I don't fully understand this sentence.

*L235-239 : it seems to me that a few models are positively biased in the Summer (ie summer Arctic sea ice extent too high), and that is can be due to sea ice still present in the nordic seas. Could it also be due to sea ice presence too further south in the Pacific sector, namely in the Bering sea? Shouldn't the authors add the Bering Sea to their "Pacific" sector?

*L245 : does it mean that the sea ice extent for all lead time equals sea ice extent at

lead time 0? It is not fully clear, as hindcasts are likely initialized near January, and not in September...

L261-263 : so what? This very interesting discussion misses a conclusion.

L297 : I don't see that.

4.Discussion and conclusion

The authors should first remind of the main conclusions of their study, then start the discussion.

L333 : underestimate -> underestimation ; overestimate -> overestimation

*L349-352 : are we sure of the causality? How is the AMOC in uninitialized simulations with CFSv2?

L350-352 : so what? Would anomaly initialization avoid such problem (instead of full initialization)?

L354 : AASW -> AABW

L363-365 : see comment above (349-352).

L425 : "but also variables (...) THAT influence ..."

L434 and sq : the authors should explain better what they mean by "direct and indirect" initialization.

L446 and sq : the authors may wish to refer to the work by Chevallier et al. (2016) on an intercomparison of Arctic sea ice cover in reanalyses, in order to illustrate that there is still some way to go.

Tables and Figures

*Table 1 : the authors should highlight clearly what models used anomaly, direct or indirect full field initialization.

*Figure 2 (7) : I have a methodological concern in figures 2 and 7 : shouldn't the authors show debiased sea ice extent for models that used anomaly initialization?

*Figure 3 : I see dashed areas on the figures that are not explained in the caption. Could the authors add a few word on that, or remove those? This dashed areas sometimes spead much southward than an acceptable model would do for September Arctic sea ice cover. . .

*Figure 4 (9) : the authors should show the 95% confidence intervals.

Figure 12 / 13 : I don't understand if the curves represent all the hindcasts, or an average over the all hindcasts. The caption ("thin gray line represents each ensemble member...") is misleading, and I am not sure there isn't any methodological error here (average over all "member 1", average over all "member 2"...?).

References

Chevallier, M. and coauthors, 2016, Intercomparison of the Arctic sea ice cover in global ocean–sea ice reanalyses from the ORA-IP project, Climate Dynamics, 1-30.

---

## Referee Comment (RC2) · Anonymous Referee #2 · 22 Jun 2016

Review of the manuscript "Assessment of Arctic and Antarctic Sea Ice Predictability in CMIP5 Decadal Hindcasts" (tc-2016-97) by C.-Y. Yang et al., submitted to The Cryosphere.

This study analyzed decadal hindcasts/predictions of Arctic and Antarctic Sea Ice from 11 CMIP5 models. The manuscript suggests that the broader prediction skill for the Arctic sea ice at increasing time leads is mainly due to the predicted decline of Arctic sea ice induced by anthropogenic forcing. In contrast, the Antarctic sea ice decadal hindcasts do not show broad predictive skill at any time scales, and almost all models predict the decline in Antarctic sea ice, opposite to the observations. The subject of the manuscript is suitable for The Cryosphere, and the results are interesting and contribute to the understanding of decadal prediction of Arctic and Antarctic sea ice. Some clarifications/diagnoses as suggested below would be helpful to strengthen the

[Figure]

manuscript. I recommend the paper to be accepted for publications in The Cryosphere with minor revisions outlined below.

1, Page 6, Lines 126-128, as mentioned here, models tend to drift away quickly from the initialized states. Will the prediction results shown in this study change if the systematic model drift is removed (i.e. drift correction) before preforming the analyses?

2, The manuscript shows that the CMIP5 decadal prediction of sea ice extent is strongly affected by anthropogenic external forcing (i.e. decline in both Arctic and Antarctica sea ice extent). How is the CMIP5 decadal prediction of Arctic and Antarctic sea ice extent compared to uninitialized CMIP5 historical+RCP4.5 simulations? Is the predictive skill enhanced with initialization compared to uninitialized hindcasts?

3, Page 18, Lines 380-387, a very recent study (Zhang, 2015) suggested that to predict September Arctic sea ice extent variations, it is important to monitor internal variability associated with the three key contributors (Atlantic/Pacific heat transport into the Arctic, and Arctic Dipole), in addition to the focus on anthropogenic changes. The study also pointed out that the Atlantic heat transport is the prime driver for low-frequency variability of winter Arctic sea ice extent, while all three contributors (Atlantic/Pacific heat transport and AD) are important for summer Arctic sea ice extent variability at low frequency. Please add discussions on these related results.

Reference: Zhang, 2015, Mechanisms for low-frequency variability of summer Arctic sea ice extent. PNAS, 112, DOI:10.1073/pnas.1422296112.

4, Almost all models predict the decline in Antarctic sea ice, opposite to the observations. Please add more discussions on what caused such a discrepancy (internal variability, ozone depletion?).

---

## Author Comment (AC1) · 13 Aug 2016

**Response to the reviews of TC-2016-97 "Assessment of Arctic and Antarctic sea ice predictability in CMIP5 decadal hindcasts" by Chao-Yuan Yang, Jiping Liu, Yongyun Hu, Radley M. Horton, Liqi Chen, Xiao Cheng**

**Response to comments by Referee #1**

**We would like to thank the reviewer for the helpful comments on the paper.**

The authors present an evaluation of sea ice covers in decadal hindcasts run with 11 models for CMIP5. They consider September Arctic and Antarctic sea ice concentration and extent, compares with observations over the period 1980-2006 (+10 years), and evaluate the ability of the models (and the multi-model ensemble mean) to capture the raw fields, the trends, regional features and anomalies relative to the trends.

Presenting an evaluation of the CMIP5 decadal experiments is a good thing per se, thus this paper is useful and timely. It actually reads well and presents interesting results. This draft shows that the predictive skill at decadal time scale is poor for both Arctic and Antarctic September sea ice extent. Yet, some skill arises in the Arctic due to the decreasing trend (anthropogenically forced) that models are able to reproduce. In the Antarctic, the lack of predictive skill is attributed to the models not being able to capture the recent increasing trend. It would have been interesting to compare predictive skill for maximum sea ice covers in both hemispheres, ie September Antarctic sea ice and March Arctic sea ice. Germe et al (2014) wrote a very interesting study on the predictability of winter Arctic sea ice based on a CMIP5 hindcast (not used in the present draft), which gave some clues on possible processes to investigate. September Arctic sea ice extent is in the spotlight, but few skill should be expected from CMIP5 decadal hindcasts besides the trend, due to the date of initialization early in the year (around January 1 for most systems I guess) and the predictability barrier documented in Day et al (2014b).

I noticed a few major points that should be addressed before possible publication: some methodological issues, some unclear figures, and a poor discussion of the results. These points are marked with an asterisk (*), and can be considered as "major comments".

**We thank the reviewer for the helpful comments. Based on your suggestions, we have conducted several additionally analyses in this response letter and modified the manuscript text accordingly. For example, with respect to the specific suggestion above, we repeated the analysis for March Arctic sea ice cover (see Figures R1-R4) and contrast it with our September result and other researchers' studies in the discussion and conclusion. Specifically, there is a smaller multi-model spread in the simulated March sea ice extent compared to that of the simulated September ice extent. As for the predictability of sea ice concentrations (anomaly correlation analysis), all the models only**

show scattered areas with significant predictive skill at the lead-time of 1-year. Unlike September, there is no significant expansion of areas with significant predictive skill as the lead-time increases. After the trend is removed, there is no much change in the areas with significant predictive skill compared to the analysis having trend. This is partly due to that the observed and simulated March sea ice extent trends are small.

We also discussed the impact of the date of initialization on the results (see response on Page 13).

[Figure]

**Figure R1 Time series of March Arctic sea ice extent (seasonal maximum) from the simulations of the 10-year hindcast for each ensemble member of each individual model (thin gray line), the ensemble mean of each individual model (thick red line) and satellite observation (black line) from 1981 to 2015.**

[Figure]

**Figure R2 Anomaly correlation coefficients between the simulated and observed Arctic March sea ice concentration anomalies for the lead-time of 1-year (top panel) and 3-5 years (bottom panel). The correlation coefficient 0.61, 0.73 and 0.88 represents 90%, 95% and 99% confidence levels, respectively. Horizontal lines depict the areas where the model simulation has sea ice whereas the observation does not have sea ice. The opposite is the case for vertical lines.**

**1 Year**

[Figure]

**3-5 Year**

**Figure R3 Same as Figure R2, but for detrended September sea ice concentration anomalies.**

[Figure]

**Figure R4 Anomaly correlation coefficients between the simulated and observed Arctic March sea ice extent anomalies for the three regional indices (the entire Arctic, Pacific and Atlantic) as a function of the lead-time. The horizontal dashed and solid lines represent 90%, 95% and 99% confidence levels, respectively. The thick gray line is the persistence prediction.**

1. Introduction

L51: "ie" -> "eg"

L51-52: the sentence sounds weird. Does it mean that sea ice treatment is important to reduced the remaining "large uncertainties for decadal climate prediction"?

**We changed the sentence to "Thus, sea ice simulation and prediction is one of the most challenging and important issues in decadal climate prediction (e.g., Meehl et al., 2009)."**

L111: "implemented an experiment" -> "implemented an experimental framework…"

**We changed the sentence to "The recent Coupled Model Intercomparison Project Phase 5 (CMIP5) has implemented an experimental framework to simulate and predict decadal climate variability…"**

L113: "validation" -> "evaluation"

**We changed "validation" to "evaluation".**

2. Models and data

*How did the authors select those 11 models? Why not using e.g. CNRM-CM5 (documented in Germe et al., 2014) or EC-Earth?

**The CMIP5 data portal provides more than 11 models having decadal experiments. The reason that 11 models were selected in this paper was that we restricted our analysis to the models with outputs of sea ice concentration from decadal hindcasts initialized every 5-years from 1981 to 2006. When we downloaded the model outputs and performed the**

analysis, not every model in the CMIP5 data portal provided decadal hindcasts initialized every 5-years from 1981 to 2006. Currently, the CMIP5 data portal (https://pcmdi.llnl.gov/projects/esgf-llnl/) does not have the monthly-mean sea ice concentration for decadal experiments for CNRM-CM5 and EC-Earth. However, we, in responding to your comments, found the Data Distribution Centre of the Intergovernmental Panel on Climate Change (IPCC, http://www.ipcc-data.org/) provides sea ice concentration for decadal experiments for CNRM-CM5, but EC-Earth has incomplete outputs (missing ensemble members). Here we repeated the analysis for CNRM-CM5. Figures R5-R14 show the results (see below). In general, the results of CNRM-CM5 are consistent with our main conclusions. CNRM-CM5, for the Arctic, show small clustered areas with significant predictive skill at the lead-time of 1-year. As the lead-time increases, the areas with significant predictive skill expand remarkably. After the trend is removed, the areas with significant predictive skill at longer time scales shrink greatly. Unlike the Arctic counterpart, there is minimal change in the areas showing significant predictive skill as the lead-time increases for the Antarctic.

**Figures for the Arctic:**

[Figure]

**Figure R5 Time series of September Arctic sea ice extent (seasonal minimum) from the simulations of the 10-year hindcast for each ensemble member of CNRM-CM5 (thin red line), the ensemble mean of CNRM-CM5 (thick red line) and satellite observation (thick black line) from 1981 to 2015.**

[Figure]

**Figure R6** Anomaly correlation coefficients between the simulated and observed Arctic September sea ice concentration anomalies of CNRM-CM5 for the lead-time of 1-year (left panel) and 3-5 years (right panel). The correlation coefficient 0.61, 0.73 and 0.88 represents 90%, 95% and 99% confidence levels, respectively. Horizontal lines depict the areas where the model simulation has sea ice whereas the observation does not have sea ice. The opposite is the case for vertical lines.

[Figure]

**Figure R7** The predicted trends (slope of a linear regression) of September Arctic sea ice extent anomalies as a function of the lead-time after applying a 3-year average. The dots represent the trend exceeding 95% confidence level.

[Figure]

**Figure R8 Same as Figure R7, but for the detrended September sea ice concentration anomalies.**

[Figure]

**Figure R9 Anomaly correlation coefficients between the simulated and observed Arctic September sea ice extent anomalies for the three regional indices (the entire Arctic, Pacific and Atlantic) as a function of the lead-time. The upper and lower panels are the original and detrended time series, respectively. The horizontal dotted, dashed and solid lines represent 90%, 95% and 99% confidence levels, respectively. The thick gray line is the persistence prediction.**

**Figures for the Antarctic:**

[Figure]

**Figure R10 Time series of September Antarctic sea ice extent (seasonal minimum) from the simulations of the 10-year hindcast for each ensemble member of CNRM-CM5 (thin red line), the ensemble mean of CNRM-CM5 (thick red line) and satellite observation (thick black line) from 1981 to 2015.**

[Figure]

**Figure R11 Anomaly correlation coefficients between the simulated and observed Antarctic September sea ice concentration anomalies for the lead-time of 1-year (top panel) and 3-5 years (bottom panel). The correlation coefficient 0.61, 0.73 and 0.88 represents 90%, 95% and 99% confidence levels, respectively. Horizontal lines depict the areas where the model simulation has sea ice whereas the observation does not have sea ice. The opposite is the case for vertical lines.**

[Figure]

**Figure R12 The predicted trends (slope of a linear regression) of September Antarctic sea ice extent anomalies as a function of the lead-time after applying a 3-year average. The dots represent the trend exceeding 95% confidence level.**

[Figure]

**Figure R13 Same as Figure R11, but for the detrended September sea ice concentration anomalies.**

[Figure]

**Figure R14 Anomaly correlation coefficients between the simulated and observed Antarctic September sea ice extent anomalies for the three regional indices (the entire Antarctic, eastern Pacific and Atlantic) as a function of the lead-time. The upper and lower panels are the original and detrended time series, respectively. The horizontal dotted, dashed and solid lines represent 90%, 95% and 99% confidence levels, respectively. The thick gray line is the persistence prediction.**

*In the first paragraph, the authors explain that there are two approaches for initialization, full-field and anomaly initialization. On L163, they introduce direct vs indirect initializations, which are not defined here.

**Thanks for the reviewer's comment. We added the following text: "As shown in Table 1, we note that four models (CanCM4, CFSv2, GEOS-5, and HadCM3) assimilate observed sea ice concentration from different resources into their sea ice initial conditions (hereafter referred to as direct sea ice initialization), whereas the rest of the models constrain sea ice initial conditions through assimilating observed ocean variables (i.e., temperature), in which the sea ice initial condition is indirectly influenced by different ocean assimilation approaches (hereafter referred to as indirect sea ice initialization). Note that the direct vs. indirect initializations are different from the aforementioned full-field vs. anomaly initializations."**

*There is no information on the start date (ie: time of the year when the hindcast are initialized). Are all models initialized in early January? Earlier? During January, the Antarctic sea ice cover is low: could it be a reason for the lack of skill (e.g. no persistence possible)?

**All the models were initialized on January 1st, except for CFSv2 and HadCM3, which**

were initialized on November 1st. We added this information into Table 1 (see below). We agree with the reviewer. We added this in the discussion: "One possibility for the lack of predictive skill for Antarctic sea ice might be that all the models were initialized on January 1st, except CFSv2 and HadCM3 (see Table 1), since the low Antarctic sea ice cover (confined to coastal Antarctica) of January 1st translates into little persistence, and little sea ice 'information'. Although CFSv2 and HadCM3 were initialized on November 1st, at a time of larger Antarctic sea ice cover, they do not show better predictability than the models initialized on January 1st. This suggests another possible explanation for the lack of predictive skill: most models cannot predict the observed increasing Antarctic sea ice in recent decades.

**TABLE1 Summary of initialization methods and data sources used for the CMIP5 decadal hindcast/prediction.**

| Model | Resolution (sea ice model) | Ensemble members | Initialization date | Sea ice assimilation method and data source |
|---|---|---|---|---|
| BCC-CM1.1 | 1 lon x 1-1/3 lat | 4 | Jan. 1st | Indirect full-field sea ice initialization (The initial sea ice indirectly influenced by nudging T to SODA ocean reanalysis) |
| CanCM4 | ~2.8 lon x 2.8 lat | 10 | Jan. 1st | Direct full-field sea ice initialization (SIC from HadISST1.1 and SIT from model-based climatology, Merryfield et al., 2013) |
| CCSM4 | 0.9 lon x 1.25 lat | 10 | Jan. 1st | Indirect full-field sea ice initialization (The initial sea ice indirectly influenced by bias-corrected CORE2-forced ocean hindcast) |
| CFSv2 | 0.5 lon x 0.5 lat | 4 | Nov. 1st | Direct full-field sea ice initialization (SIC from NCEP climate forecast system reanalysis) |
| FGOALS-g2 | 1 lon x 1 lat | 3 | Jan. 1st | Indirect full-field sea ice initialization (The initial sea |

| | | | | |
|---|---|---|---|---|
| | | | | ice indirectly influenced by nudging T and S to an ocean reanalysis) |
| GEOS-5 | 1 lon x 1 lat | 3 | Jan. 1st | Direct full-field sea ice initialization (SIC from GEOS-iODAS) |
| GFDL-CM2.1 | ~1 lon x 0.75 lat | 10 | Jan. 1st | Indirect full-field sea ice initialization (The initial sea ice indirectly influenced by atmospheric and ocean data, Msadek et al. 2014) |
| HadCM3 | 1.25 lon x 1.25 lat | 10 | Nov. 1st | Direct anomaly sea ice initialization (SIC from Met Office Hadley Centre sea ice data, HadISST) |
| IPSL-CM5A-LR | ~2 lon x 2 lat | 6 | Jan. 1st | Indirect anomaly sea ice initialization (The initial sea ice indirectly influenced by the assimilation of T and S anomalies from observations) |
| MIROC5 | 1 lon x 1 lat | 6 | Jan. 1st | Indirect anomaly sea ice initialization (The initial sea ice is indirectly influenced by the assimilation of T and S from an objective analysis of Ishii and Kimoto, 2009) |
| MPI-ESM-MR | ~0.4 lon x 0.4 lat | 3 | Jan. 1st | Indirect anomaly sea ice initialization (The initial sea ice indirectly influenced by the assimilation of T and S anomalies from a forced ocean run using NCEP reanalysis, Müller et al., 2012) |

L139-141: Let's be honest: the main reason why we don't focus on the Antarctic sea ice minimum is because we know the models poorly simulate this minimum. See for instance Turner et al., 2013.

**We modified the text, with item '2' reflecting your exact point, thanks:   "…1) sea ice in the Antarctic largely melts away (confined to coastal Antarctica) during the seasonal minimum (i.e. February or March), 2) previous studies (e.g., Turner et al., 2013; Meijers, 2014) have shown that climate models poorly simulate seasonal minimum, and 3) September sea ice extent has a significant increasing trend."**

*Meijers, A., The Southern Ocean in the Coupled Model Intercomparison Project phase 5. Philos. Trans. Roy. Soc., 372A, doi:10.1098/rsta.2013.0296, 2014.*

3. Prediction skill of CMIP5 decadal hindcasts

*A discussion on the significance of the predicted trends is missing, as those trends seem to be calculated based on only 6 points (this is actually not well explained)…

**Based on your suggestions, in this revision, we added a significance test for the predicted trends in the figures (see below Figure R15 and R16, which are included in the manuscript as Figures 4 and 9) and more discussion about the trends. The dots in these figures represent the trends exceeding 95% confidence levels.**

[Figure]

**Figure R15 The predicted trend (slope of a linear regression) of September Arctic sea ice extent anomalies as a function of lead times after applying a 3-year average to filter out high frequency variability. The dots represent the trends exceeding 95% confidence levels**

[Figure]

**Figure R16 The predicted trend (slope of a linear regression) of September Antarctic sea ice extent anomalies as a function of lead times after applying a 3-year average to filter out high frequency variability. The dots represent the trends exceeding 95% confidence levels.**

L163: see above: what is "direct and indirect full initialization"?

**We added descriptions for direct and indirect initializations in "Models and Data" section (see our response above and L142-148 in modified manuscript).**

L165: what is "improved initialization"? Compared to what?

**We changed the sentence to "Hence initializations that with values from various best estimates of sea ice state do not necessarily mitigate drift…"**

L172-174: the formula is wrong. Pbar and Obar should be divided by n to denote an average.

**Thanks. We corrected the formula to**

$$ACC = \frac{\sum_{i=1}^{n}[P(i,t) - \overline{P}(t)] \cdot [O(i,t) - \overline{O}(t)]}{\sqrt{\sum_{i=1}^{n}[P(i,t) - \overline{P}(t)]^2 \cdot \sum_{i=1}^{n}[O(i,t) - \overline{O}(t)]^2}}$$

**where P is the predicted sea ice concentration and $\overline{P}(t)$ is calculated as $\sum_{i=1}^{n} P(i,t)/n$; O is the observed sea ice concentration and $\overline{O}(t)$ is calculated as $\overline{O}(t) = \sum_{i=1}^{n} O(i,t)/n$**

L200-202: the contradiction in this sentence strikes me.

**We changed the sentence to "In general, the MMEM has better predictive skill than the**

**majority of the models for all lead-times.”**

L210/212: "lead-time" or "lead time"? The authors should be consistent.

**Thanks for the reviewer's comment. For consistency, we use "lead-time" throughout the paper.**

L212-213: "those longer lead times are weighted towards inclusion of more recent years in the observations with accelerated decline of Arctic sea ice". I don't fully understand this sentence.

**We improved the writing. What we meant was that longer lead-times, by definition, exclude more early years that the downward trends were smaller than they have been in recent years. For example, Figure R17 shows the time-series of observed sea ice extent (dots) and corresponding trend (solid line) for the lead-time of 1 year (red) and 7 year (blue), respectively. Clearly, the time series with more data points in recent years (blue) has larger negative trend. We changed the sentence to "…those longer lead times are weighted towards inclusion of more data points in recent years, the years with accelerated decline of Arctic sea ice."**

[Figure]

**Figure R17 The timeseries of observed sea ice extent (dots) and corresponding trend (solid line) for lead-time of 1 year (red) and 7 year (blue), respectively.**

*L235-239: it seems to me that a few models are positively biased in the Summer (ie summer Arctic sea ice extent too high), and that is can be due to sea ice still present in the nordic seas. Could it also be due to sea ice presence too further south in the Pacific sector, namely in the Bering sea? Shouldn't the authors add the Bering Sea to their "Pacific" sector?

**We re-plotted Figure 3 with spatial coverage extended to 50N. In Figure R18 (the replotted Figure 3), horizontal lines mean the areas where the model simulation has sea**

ice whereas the observation does not have sea ice. The opposite is the case for vertical lines. For the Pacific sector of the Arctic, perhaps interestingly there is no sea ice in the Bering Sea in September in any of the model simulations, which is the reason that we did not include the Bering Sea to generate the regional index. For the models that are positively biased in September, they simulate more sea ice in the Barents Sea (Figure R18 vs. Figure 2). We added the following text: "In Figure 3, horizontal lines mean the areas where the model simulation has sea ice whereas the observation does not have sea ice. The opposite is the case for vertical lines. For the Pacific sector of the Arctic, there is no sea ice in the Bering Sea in September for any model simulations. The models that are positively biased in September simulate too much sea ice in the Barents Sea."

[Figure]

**Figure R18 Anomaly correlation coefficients between the simulated and observed Arctic September sea ice concentration anomalies for the lead-time of 1-year (top panel) and 3-5 years (bottom panel). The correlation coefficient 0.61, 0.73 and 0.88 represents 90%, 95% and 99% confidence levels, respectively. Horizontal lines depict the areas where the model simulation has sea ice whereas the observation does not have sea ice. The opposite is the case for vertical lines.**

*L245: does it mean that the sea ice extent for all lead time equals sea ice extent at lead time 0? It is not fully clear, as hindcasts are likely initialized near January, and not in September…

**Yes, the persistence prediction means that the predicted sea ice extent for all lead-time equals the ice extent at lead-time 0. We used this to generate persistence prediction every**

**5-year from 1981 to 2006. For example, for the Arctic, we used the simulated September sea ice extent in the year of 1981 from the first 10-year hindcast, 1986 from the second 10-year hindcast… as lead-time 0. In the persistence case, the predicted sea ice extent for lead-time 1, i.e., 1982 from the first 10-year hindcast, 1987 from the second 10-year hindcast… by definition has the same ice extent as lead-time 0.**

L261-263: so what? This very interesting discussion misses a conclusion.

**We added a conclusion here: "Additionally, all the models, except CanCM4, appear to have a re-emerging predictive skill for the north Atlantic sea ice after 6-8 years (Fig. 6e). This might be associated with the influence of Atlantic meridional overturning circulation (see Discussion for details)."**

L297: I don't see that.

**We changed the sentence to "After the linear trend is removed, the areas having significant predictive skill become relatively broader for the majority of the models compared to those of the raw data at longer lead-times, i.e., the MMEM has relatively better predictive skill in the northern Ross Sea and a large portion of the Weddell Sea at the lead-time of 3-5 years (Fig. 10 vs. Fig. 8)."**

4. Discussion and conclusion
The authors should first remind of the main conclusions of their study, then start the discussion.

**Thanks for your suggestions. We re-organized the "Discussion and conclusion" section by first reminding of the main conclusions of our study (see modified manuscript on page 15-23)**

L333: underestimate ->underestimation; overestimate -> overestimation

**We changed the sentence to "…not only the underestimation of observed September sea ice cover, but also the overestimation of observed March sea ice cover…"**

*L349-352: are we sure of the causality? How is the AMOC in uninitialized simulations with CFSv2?

**This is a good question. Unfortunately, for CMIP5, CFSv2 did not provide uninitialized simulations, including historical and pre-industrial simulations. Thus, we are not able to**

**analyze the AMOC in uninitialized CFSv2 simulations. Here we calculated the ensemble mean stream function for the CFSv2 decadal hindcasts. It appears that the AMOC is weak even at the beginning of the simulation for all hindcasts (~6-12Sv) compared to the observation (18.7Sv in Cunningham et al., 2007; 17.2Sv in Smeed et al., 2014; McCarthy et al., 2015), and decreases substantially during the decadal hindcast. Hence, the AMOC in CFSv2 might not be able to transport necessary warm water into the Arctic Ocean, leading to the excessive sea ice melt.**

L350-352: so what? Would anomaly initialization avoid such problem (instead of full initialization)?

**This is a good question but CFSv2 did not provide decadal simulations using anomaly initialization. Hence we cannot investigate whether anomaly initialization avoids this problem. We will investigate this issue in future research.**

L354: AASW -> AABW

**Thanks, we corrected it.**

L363-365: see comment above (349-352).

**Like CFSv2, GEOS-5 also did not provide uninitialized simulations, including historical, and pre-industrial simulations. Hence we cannot investigate this further, but we now note that it is an important area for future research/model experiments.**

L425: "but also variables (…) THAT influence…"

**We changed the sentence to "…but also variables (i.e., sea ice thickness) that influence surface energy fluxes and, thereby, ocean-atmosphere interaction."**

L434 and sq: the authors should explain better what they mean by "direct and indirect" initialization.

**Thanks. We added descriptions for direct and indirect initializations in "Models and Data" section (see our response above).**

L446 and sq: the authors may wish to refer to the work by Chevallier et al. (2016) on an intercomparison of Arctic sea ice cover in reanalyses, in order to illustrate that there is still

some way to go.

**Thanks for the reviewer's suggestion. We added this: "A recent study (Chevallier et al., 2016) showed that global reanalyses that do not assimilate sea ice concentration generally overestimate sea ice concentration and have large biases near the ice edge in the Arctic. They also pointed out that none of global reanalyses has assimilated sea ice thickness data. Thus, efforts should be devoted to further development of initialization…"**

*Chevallier, M., Smith, G., Lemieux, J.-F., Dupont, F., Forget, G., Fujii, Y., Hernandez, F., Msadek, R., Peterson, K.A., Storto, A., Toyoda, T., Valdivieso, M., Vernieres, G., Zuo, H., Balmaseda, M., Chang, Y.-S., Ferry, N., Garric, G., Haines, K., Keeley, S., Kovach, R.M., Kuragano, T., Masina, S., Tang, Y., Tsujino, H., Wang, X. Intercomparison of the Arctic sea ice cover in global ocean-sea ice reanalyses from the ORA-IP project. Climate Dynamics, Special Issue: Ocean Reanalysis, online, doi:10.1007/s00382-016-2985-y, 2016.*

Tables and Figures
*Table 1: the authors should highlight clearly what models used anomaly, direct or indirect full field initialization.

**Thanks. We added this information in Table 1 (see our response above).**

*Figure 2 (7): I have a methodological concern in figures 2 and 7: shouldn't the authors show debiased sea ice extent for models that used anomaly initialization?

**The main purpose of this paper is to show the current status of sea ice prediction in the Arctic and Antarctic on decadal timescales for current-day coupled global climate models. This is why we did not remove the model bias for the models. Here we used the bias-correction method mentioned in Ham et al. (2014) to remove the model drift (see Figure R19 and R20). This method removes the lead-time dependent mean bias based on the observation. The bias-corrected decadal hindcast is calculated as:**

$$\widehat{Y_{jt}} = Y_{jt} - \sum_{k=1}^{N}(Y_{kt} - O_{kt})/N$$

**where $Y_{jt}$ and $\widehat{Y_{jt}}$ are the raw and bias-corrected predicted sea ice state, at the initialized year j and lead year t. $O_{jt}$ is the observed sea ice state. We also applied this method to re-calculate the anomaly correlation coefficient between the observed and bias-corrected simulated regional sea ice indices for the Arctic and Antarctic (see Figure R21 and R22). The results for the bias-corrected decadal hindcasts are similar to those having systematic model drift (Figure R21(R22) vs. Figure 6(11)). This is because the bias**

correction only minimally influences the variability of the time-series as reflected by the anomaly correlation coefficient.

*Ham, Y.-G., Rienecker, M. M., Suarez, M. J., Vikhliaev, Y., Zhao, B., Marshak, J., Vernieres, G., and Schubert, S. D., Decadal prediction skill in the GEOS-5 forecast system. Clim. Dyn., 42, 1-20, 2014.*

[Figure]

**Figure R19 Time series of September Arctic sea ice extent (seasonal minimum) from the simulations of the 10-year hindcast for the ensemble mean of each individual model (thick red line), the bias-corrected ensemble mean of each individual model (thick blue line) and satellite observation (black line) from 1981**

to 2015.

[Figure]

**Figure R20 Time series of September Antarctic sea ice extent (seasonal maximum) from the simulations of the 10-year hindcast for the ensemble mean of each individual model (thick red line), the bias-corrected ensemble mean of each individual model (thick blue line) and satellite observation (black line) from 1981 to 2015.**

[Figure]

**Figure R21 Anomaly correlation coefficients between the bias-corrected simulated and observed Arctic September sea ice extent anomalies for the three regional indices (the entire Arctic, Pacific and Atlantic) as a function of the lead-time. The top and bottom panels are the original and detrended time series, respectively. The horizontal dotted, dashed and solid lines represent 90%, 95% and 99% confidence levels, respectively. The thick gray line is the persistence prediction.**

[Figure]

**Figure R22 Anomaly correlation coefficients between the bias-corrected simulated and observed Antarctic September sea ice extent anomalies for the three regional indices (the entire Antarctic, eastern Pacific and Atlantic) as a function of the lead-time. The top and bottom panels are the original and detrended time series, respectively. The horizontal dotted, dashed and solid lines represent 90%, 95% and 99% confidence levels, respectively. The thick gray line is the persistence prediction.**

*Figure 3: I see dashed areas on the figures that are not explained in the caption. Could the authors add a few word on that, or remove those? This dashed areas sometimes spread much southward than an acceptable model would do for September Arctic sea ice cover: : :

**We re-plotted Figure 3 and changed the caption to "Figure 3. Anomaly correlation coefficients between the simulated and observed Arctic September sea ice concentration anomalies for the lead-time of 1-year (top panel) and 3-5 years (bottom panel). The correlation coefficient 0.61, 0.73 and 0.88 represents 90%, 95% and 99% confidence levels, respectively. Horizontal lines depict the areas where the model simulation has sea ice whereas the observation does not have sea ice. The opposite is the case for vertical lines."**

*Figure 4 (9): the authors should show the 95% confidence intervals.

**Thanks for the reviewer's comment. We added significant test for the predicted trends in the figures (see above Figure R15 and R16).**

Figure 12 / 13: I don't understand if the curves represent all the hindcasts, or an average over the all hindcasts. The caption ("thin gray line represents each ensemble member...") is misleading, and I am not sure there isn't any methodological error here (average over all "member 1", average over all "member 2"...?).

**As shown in Table 1, each CFSv2 decadal hindcast has four ensemble members. The thick black line in the figure represents the average for the four ensemble members as a function of lead-time. The gray line represents each individual ensemble member.**

References
Chevallier, M. and coauthors, 2016, Intercomparison of the Arctic sea ice cover in global ocean–sea ice reanalyses from the ORA-IP project, Climate Dynamics, 1-30.

**We cited it in the paper.**

---

## Author Comment (AC2) · 13 Aug 2016

**Response to the reviews of TC-2016-97 "Assessment of Arctic and Antarctic sea ice predictability in CMIP5 decadal hindcasts" by Chao-Yuan Yang, Jiping Liu, Yongyun Hu, Radley M. Horton, Liqi Chen, Xiao Cheng**

**Response to comments by Referee #2**

**We would like to thank the reviewer for the helpful comments on the paper.**

This study analyzed decadal hindcasts/predictions of Arctic and Antarctic Sea Ice from 11 CMIP5 models. The manuscript suggests that the broader prediction skill for the Arctic sea ice at increasing time leads is mainly due to the predicted decline of Arctic sea ice induced by anthropogenic forcing. In contrast, the Antarctic sea ice decadal hindcasts do not show broad predictive skill at any time scales, and almost all models predict the decline in Antarctic sea ice, opposite to the observations. The subject of the manuscript is suitable for The Cryosphere, and the results are interesting and contribute to the understanding of decadal prediction of Arctic and Antarctic sea ice. Some clarifications/diagnoses as suggested below would be helpful to strengthen the manuscript. I recommend the paper to be accepted for publications in The Cryosphere with minor revisions outlined below.

1. Page 6, Lines 126-128, as mentioned here, models tend to drift away quickly from the initialized states. Will the prediction results shown in this study change if the systematic model drift is removed (i.e. drift correction) before performing the analyses?

**Thanks for the reviewer's comment. Here we used the bias-correction method mentioned in Ham et al. (2014) to remove the model drift (see Figure R1 and R2). This method removes the lead-time dependent mean bias based on the observation. The bias-corrected decadal hindcast is calculated as:**

$$\widehat{Y_{jt}} = Y_{jt} - \sum_{k=1}^{N} (Y_{kt} - O_{kt})/N$$

**where $Y_{jt}$ and $\widehat{Y_{jt}}$ are the raw and bias-corrected predicted sea ice state, at the initialized year j and lead year t. $O_{jt}$ is the observed sea ice state. We also applied this method to re-calculate the anomaly correlation coefficient between the observed and bias-corrected simulated regional sea ice indices for the Arctic and Antarctic (Figure R3 and R4). The results for the bias-corrected decadal hindcasts are similar to those having systematic model drift (Figure R3(R4) vs. Figure 6(11)). This is because the bias correction only minimally influences the variability of the time-series as reflected by the anomaly correlation coefficient.**

*Ham, Y.-G., Rienecker, M. M., Suarez, M. J., Vikhliaev, Y., Zhao, B., Marshak, J., Vernieres, G., and Schubert, S. D., Decadal prediction skill in the GEOS-5 forecast system. Clim. Dyn., 42, 1-20, 2014.*

[Figure]

**Figure R1 Time series of September Arctic sea ice extent (seasonal minimum) from the simulations of the 10-year hindcast for the ensemble mean of each individual model (thick red line), the bias-corrected ensemble mean of each individual model (thick blue line) and satellite observation (black line) from 1981 to 2015.**

[Figure]

**Figure R2 Time series of September Antarctic sea ice extent (seasonal maximum) from the simulations of the 10-year hindcast for the ensemble mean of each individual model (thick red line), the bias-corrected ensemble mean of each individual model (thick blue line) and satellite observation (black line) from 1981 to 2015.**

[Figure]

**Figure R3 Anomaly correlation coefficients between the bias-corrected simulated and observed Arctic September sea ice extent anomalies for the three regional indices (the entire Arctic, Pacific and Atlantic) as a function of the lead-time. The top and bottom panels are the original and detrended time series, respectively. The horizontal dotted, dashed and solid lines represent 90%, 95% and 99% confidence levels, respectively. The thick gray line is the persistence prediction.**

[Figure]

**Figure R4 Anomaly correlation coefficients between the bias-corrected simulated and observed Antarctic September sea ice extent anomalies for the three regional indices (the entire Antarctic, eastern Pacific and Atlantic) as a function of the lead-time. The top and bottom panels are the original and detrended time series, respectively. The horizontal dotted, dashed and solid lines represent 90%, 95% and 99% confidence levels, respectively. The thick gray line is the persistence prediction.**

2. The manuscript shows that the CMIP5 decadal prediction of sea ice extent is strongly affected by anthropogenic external forcing (i.e. decline in both Arctic and Antarctica sea ice extent). How is the CMIP5 decadal prediction of Arctic and Antarctic sea ice extent compared to uninitialized CMIP5 historical+RCP4.5 simulations? Is the predictive skill enhanced with initialization compared to uninitialized hindcasts?

**As suggested by the reviewer, we downloaded the historical and RCP4.5 simulations (hereafter referred to as uninitialized simulation) for all the models except CFSv2 and GEOS-5 (they did not provide historical and RCP4.5 simulations), and then repeated the analyses. Figures R5 to R12 show the results for the uninitialized simulation. For the Arctic, the predictive skill of sea ice cover is enhanced for the initialized hindcast compared to the uninitialized simulation for most models and MMEM. After the trend is removed (Figure R8), there is no obvious difference between the initialized hindcast and the uninitialized simulation. Note that Figure R6 and R8 do not include CFSv2 and GEOS-5, which have poor predictive skills in the initialized hindcast. It is possible that the predictive skill of MMEM for the uninitialized simulation would be worse when**

**CFSv2 and GEOS-5 were included. For the Antarctic, there is no significant difference between the initialized hindcast and the uninitialized simulation, largely due to that most models in the uninitialized simulation cannot capture the observed increasing Antarctic sea ice. However, after the linear trend is removed, the areas with significant predictive skill for the initialized hindcast become relatively larger relative to those of the uninitialized simulation (Figure R10 and R12).**

**Figures for the Arctic:**

[Figure]

**Figure R5 Time series of September Arctic sea ice extent (seasonal minimum) from the simulations of the historical scenario for each individual model (thick blue line), the simulations of the rcp45 scenario for each individual model (thick red line) and satellite observation (black line) from 1981 to 2015.**

[Figure]

**Figure R6 Anomaly correlation coefficients between the simulated and observed Arctic September sea ice concentration anomalies for the lead-time of 1-year (left panel) and 3-5 years (right panel). The correlation coefficient 0.61, 0.73 and 0.88 represents 90%, 95% and 99% confidence levels, respectively. Horizontal lines depict the areas where the model simulation has sea ice whereas the observation does not have sea ice. The opposite is the case for vertical lines.**

[Figure]

**Figure R7 The predicted trend (slope of a linear regression) of September Arctic sea ice extent anomalies as a function of lead times after applying a 3-year average to filter out high frequency variability. The dots represent the trend exceeding 95% confidence level.**

[Figure]

**Figure R8 Same as Figure R6, but for detrended September sea ice concentration anomalies.**

**Figures for the Antarctic:**

[Figure]

**Figure R9 Time series of September Antarctic sea ice extent (seasonal maximum) from the simulations of the historical scenario for each individual model (thick blue line), the simulations of the rcp45 scenario for each individual model (thick red line) and satellite observation (black line) from 1981 to 2015.**

[Figure]

**Figure R10 Anomaly correlation coefficients between the simulated and observed Antarctic September sea ice concentration anomalies for the lead-time of 1-year (left panel) and 3-5 years (right panel). The correlation coefficient 0.61, 0.73 and 0.88 represents 90%, 95% and 99% confidence levels, respectively. Horizontal lines depict the areas where the model simulation has sea ice whereas the observation does not have sea ice. The opposite is the case for vertical lines.**

[Figure]

**Figure R11 The predicted trend (slope of a linear regression) of September Antarctic sea ice extent anomalies as a function of lead times after applying a 3-year average to filter out high frequency variability. The dots represent the trend exceeding 95% confidence level.**

[Figure]

**Figure R12 Same as Figure R10, but for detrended September sea ice concentration anomalies.**

3. Page 18, Lines 380-387, a very recent study (Zhang, 2015) suggested that to predict September Arctic sea ice extent variations, it is important to monitor internal variability associated with the three key contributors (Atlantic/Pacific heat transport into the Arctic, and Arctic Dipole), in addition to the focus on anthropogenic changes. The study also pointed out that the Atlantic heat transport is the prime driver for low-frequency variability of winter Arctic sea ice extent, while all three contributors (Atlantic/Pacific heat transport and AD) are important for summer Arctic sea ice extent variability at low frequency. Please add discussions on these related results.

Reference: Zhang, 2015, Mechanisms for low-frequency variability of summer Arctic sea ice extent. PNAS, 112, DOI:10.1073/pnas.1422296112.

**Thanks for the reviewer's suggestion. We added the following text in the discussion and conclusion: "Zhang (2015) suggested that it is important to monitor internal variability associated with the heat transport into the Arctic from the Atlantic and Pacific, and the Arctic Dipole for predicting September Arctic sea ice extent variations. This study also pointed out that all these processes are important for low-frequency variability of summer sea ice extent, while the Atlantic heat transport might be the prime driver for winter Arctic sea ice extent variability at low frequency"**

4. Almost all models predict the decline in Antarctic sea ice, opposite to the observations. Please add more discussions on what caused such a discrepancy (internal variability, ozone depletion?).

**Thanks for the reviewer's comment. We added the following text in the discussion and conclusion: "The reasons behind recent increase of Antarctic sea ice are complex, and several recent studies show that scientists are still trying to understand it. The possible mechanisms include variations in atmospheric circulation linked to the Antarctic Oscillation, Amundsen Sea low pressure system, stratospheric ozone depletion, and increased greenhouse gases, changes in zonal and meridional near surface winds, the increase in fresh water flux which stabilizes the upper ocean layer, and the influence of internal variability (e.g., Zhang, 2007; Turner et al., 2009; Sigmond and Fyfe, 2010; Liu and Curry 2010; Holland and Kwok, 2012; Zunz et al. 2013; Polvani and Smith, 2013). However, it is not clear which is the dominant process. Further investigating a range of other variables such as simulated sea ice thickness, sea ice velocity, near surface wind, and ocean stratification will help elucidate the reasons why the trends in these models are different from observations."**

*Holland, P. R., and Kwok, R., Wind-driven trends in Antarctic sea-ice drift, Nat. Geosci., 5, 872–875, 2012.*
*Liu, J., and Curry, J. A., Accelerated warming of the Southern Ocean and its impacts on the hydrological cycle and sea ice. Proc. Natl. Acad. Sci. USA, 107, 14 987–14 992, 2010.*
*Polvani, L. M., and Smith, K. L., Can natural variability explain observed Antarctic sea ice trends? New modeling evidence from CMIP5, Geophys. Res. Lett., 40, 3195–3199, doi:10.1002/grl.50578, 2013.*
*Sigmond, M., and Fyfe, J. C., Has the ozone hole contributed to increased Antarctic sea ice extent?, Geophys. Res. Lett., 37, L18502, doi:10.1029/2010GL044301, 2010*

*Turner, J., and Coauthors, Non-annular atmospheric circulation change induced by stratospheric ozone depletion and its role in the recent increase of Antarctic sea ice extent. Geophys. Res. Lett., 36, L08502, doi:10.1029/2009GL037524, 2009.*

*Zhang, J., Increasing Antarctic sea ice under warming atmospheric and oceanic conditions, J. Clim., 20, 2515–2529, 2007.*

*Zunz, V., H. Goosse, and Massonnet, F., How does internal variability influence the ability of CMIP5 models to reproduce the recent trend in Southern Ocean sea ice extent? Cryosphere, 7, 451–468, 2013.*